# Towards Bridging the Gap between Large-Scale Pretraining and Efficient Finetuning for Humanoid Control

**Weidong Huang**[1], **Zhehan Li**[1,2], **Hangxin Liu**[1], **Biao Hou**[2], **Yao Su**[1], **Jingwen Zhang**[1*]
[1]State Key Laboratory of General Artificial Intelligence, BIGAI
[2]School of Artificial Intelligence, Xidian University

## ABSTRACT

Reinforcement Learning (RL) is widely used for humanoid control, with on-policy methods such as Proximal Policy Optimization (PPO) enabling robust training via large-scale parallel simulation and, in some cases, zero-shot deployment to real robots. However, the low sample efficiency of on-policy algorithms limits safe adaptation to new environments. Although off-policy RL and model-based RL have shown improved sample efficiency, the gap between large-scale pretraining and efficient finetuning on humanoids still exists. In this paper, we find that off-policy Soft Actor-Critic (SAC), with large-batch update and a high Update-To-Data (UTD) ratio, reliably supports large-scale pretraining of humanoid locomotion policies, achieving zero-shot deployment on real robots. For adaptation, we demonstrate that these SAC-pretrained policies can be finetuned in new environments and out-of-distribution tasks using model-based methods. Data collection in the new environment executes a deterministic policy while stochastic exploration is instead confined to a physics-informed world model. This separation mitigates the risks of random exploration during adaptation while preserving exploratory coverage for improvement. Overall, the approach couples the wall-clock efficiency of large-scale simulation during pretraining with the sample efficiency of model-based learning during fine-tuning. **Code and videos:** `https://lift-humanoid.github.io`

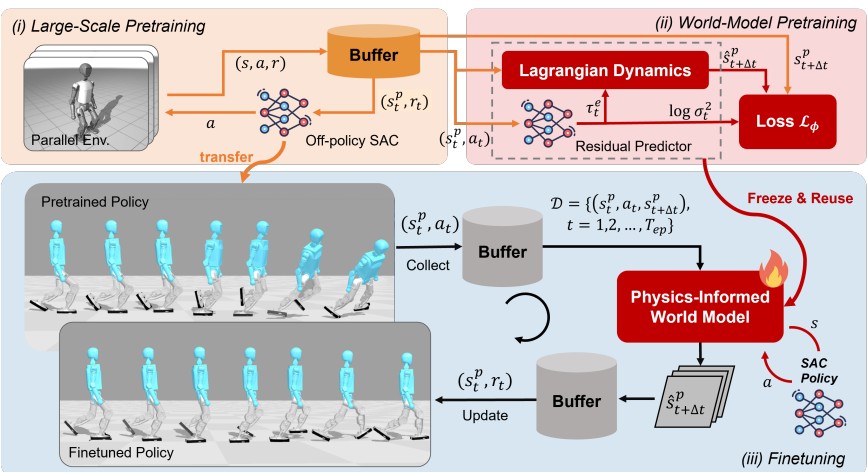

Figure 1: **L**arge-scale pretra**I**ning and efficient **F**ine**T**uning (LIFT) Framework. In stage (i), we implement SAC in JAX to support large-batch update and high UTD, achieving fast, robust convergence in massively parallel simulation and zero-shot deployment to a real humanoid in outdoor experiments. In stage (ii), we pretrain a physics-informed world model on the SAC data, combining Lagrangian dynamics with a residual predictor to capture contact forces and other unmodeled effects. In stage (iii), we finetune both the policy and the world model to new environments while executing only deterministic actions in the environment. Stochastic exploration is confined to rollouts within the world model. This framework enhances both the safety and efficiency of finetuning.

---

*Corresponding Author.

# 1 INTRODUCTION

Learning generalist physical agents is a long-standing goal in AI, and humanoid robots offer broad task coverage among robotic embodiments. PPO (Schulman et al., 2017) becomes a mainstream baseline for humanoid robot control due to its robustness and fast wall-clock convergence in massively parallel GPU simulation (Schwarke et al., 2025). However, when only limited real-world or cross-task data can be collected, on-policy methods are disadvantaged because they discard off-policy experience (Haarnoja et al., 2018b; Lillicrap et al., 2015; Fujimoto et al., 2018). Moreover, even when a PPO policy achieves zero-shot deployment in new environments, performance metrics can degenerate (Gu et al., 2024) (e.g., poorer velocity tracking or reduced execution precision). These limitations highlight the need for frameworks that can reuse prior experience and adapt policies efficiently in new environments.

Off-policy algorithms offer an appealing backbone for this paradigm because they can leverage replayed experience for sample-efficient finetuning (Lillicrap et al., 2015). However, these algorithms tend to overfit to replayed data, which can bias exploration during fine-tuning, especially when large-scale parallel simulation with them has received limited attention. During fine-tuning, directly adapting a policy to a new environment often leads to unsafe actions because of stochastic exploration. This risk is especially high for humanoid robots: small support polygons, particularly during single-support phases, make the system highly sensitive to perturbations.

Constraining exploration within a learned world model provides a safer and more sample-efficient alternative (Levy et al., 2024; Hafner et al., 2023), though synthetic rollouts may introduce model bias that destabilizes finetuning (Ha & Schmidhuber, 2018; Janner et al., 2019). A physics-informed world model (Levy et al., 2024) that incorporates structural priors from physcis improves the fidelity of synthetic rollouts, reduces model bias, and enables more reliable finetuning even with limited data. However, training such model-based systems from scratch (Levy et al., 2024) remains extremely slow in wall-clock time and prone to local minima. Meanwhile, we find that off-policy SAC (Haarnoja et al., 2018b) integrates more naturally with model-based algorithms (Levy et al., 2024) than PPO, achieving robust and sample-efficient improvements. These findings further motivate our pipeline. We introduce **L**arge-scale pretra**I**ning and efficient **F**ine**T**uning (**LIFT**), a three-stage framework as shown in the Figure 1: (i) large-scale policy pretraining, (ii) physics-informed world model pretraining, and (iii) efficient finetuning of the policy and world model. We make the following contributions:

1. We provide a scalable JAX implementation of SAC that supports robust convergence in massively parallel simulation and zero-shot deployment to a physical humanoid robot within one hour of wall-clock training on a single NVIDIA RTX 4090 GPU. The resulting SAC policy also serves as the policy module within our model-based finetuning stage.

2. We develop a fine-tuning strategy that executes deterministic actions in new environments while limiting stochastic exploration in a physics-informed world model, which improves sample efficiency and stabilizes convergence. This physics-informed design is pivotal for finetuning, enabling data-efficient in-distribution adaptation and stronger out-of-distribution generalization.

3. We release an open-source pipeline for humanoid control spanning pretraining, zero-shot deployment, and finetuning, providing a practical baseline for the robotics community.

# 2 RELATED WORKS

**Large-Scale RL Pretraining in Simulation.** With the rapid progress of GPU technology, several GPU-based physics simulators have emerged, such as IsaacGym (Makoviychuk et al., 2021), Mujoco Playground (Zakka et al., 2025), and Brax (Freeman et al., 2021). These frameworks enable fast parallel simulation of thousands of robot environments directly on GPUs. PPO (Schulman et al., 2017) has become the dominant baseline in this setting due to the ease of implementation and robust convergence. By leveraging domain randomization (Tobin et al., 2017), PPO policies can achieve fast wall-clock convergence and even zero-shot transfer to physical humanoid robots (Rudin et al., 2022; Gu et al., 2024). However, the low sample efficiency of on-policy methods limits their ability to adapt or continue training in new environments.

**Off-Policy RL.** When data collection is expensive or risky, improving sample efficiency becomes critical. Off-policy algorithms such as TD3 (Fujimoto et al., 2018) and SAC (Haarnoja et al., 2018b) achieve higher sample efficiency by reusing past experiences and leveraging critic gradients to update the policy. SAC has been used to train quadruped robots from scratch with limited real-world data (Haarnoja et al., 2018c; Ha et al., 2020; Haarnoja et al., 2018a; Smith et al., 2022b), and pretrained SAC policies can be finetuned on hardware with the same algorithm (Smith et al., 2022a; Ball et al., 2023). However, these methods typically rely on stochastic exploration in the environment, where injected action noise may damage actuators or induce unsafe states. This risk is amplified for humanoids, whose smaller support polygons (especially in single-support phases) make them sensitive to perturbations compared to quadrupeds. These methods also underutilize large-scale parallel simulation, resulting in high wall-clock training time. Recent large-scale off-policy efforts include Parallel Q-Learning (Li et al., 2023), which scales DDPG (Lillicrap et al., 2015) across massive simulations yet without sim-to-real validation, and FastTD3 (Seo et al., 2025), which achieves humanoid sim-to-real but its finetuning ability in new environments remains unclear. Raffin (2025) show that SAC can be made stable in massively parallel simulators through extensive parameter tuning, but their work also lacks sim-to-real evaluation and does not examine fine-tuning performance.

**Model-Based Techniques** Model-based methods can further improve sample efficiency, such as MBPO (Janner et al., 2019) using synthetic rollouts and Dreamer (Hafner et al., 2023) showing strong data efficiency in real-world tasks (Wu et al., 2023). However, they often still require stochastic exploration in the environment. Sun et al. (2025) train a world model that explicitly reconstructs the environment state and use it as policy input to enhance locomotion robustness, but the model is not used to generate synthetic data for policy fine-tuning aimed at improving adaptability. ASAP (He et al., 2025) collects real-world data using a pretrained policy and learns a delta-action model to correct simulator actions before fine-tuning. However, the delta network can output unbounded action corrections while finetuning, which may lead to undesired behaviors, and the approach requires substantial real-world data to sufficiently cover the gaps between the simulator and the physical robot, especially across diverse motions. These limitations highlight an open question: how can physics priors be incorporated into world models to improve predictive accuracy while reducing the amount of data required for policy fine-tuning? Physics-Informed Neural Networks (PINNs) (Raissi et al., 2019) represent one possible approach by embedding physical constraints directly into the learning architecture. For example, Greydanus et al. (2019) proposed Hamiltonian neural networks that learn energy-conserving dynamics in an unsupervised manner, while Cranmer et al. (2020) introduced Lagrangian neural networks that model dynamics without requiring canonical coordinates. Deep Lagrangian Networks (Lutter et al., 2019) further demonstrated strong performance in robot tracking and extrapolation to novel trajectories. Yet these approaches remain limited to energy-conserving systems and struggle with contact-rich dynamics. To address this, SSRL (Levy et al., 2024) proposed combining rigid-body dynamics with learned contact-force models, enabling a quadruped robot to learn to walk with just three minutes of real-world data. Nevertheless, training from scratch remains unsafe for humanoids due to their fragility and instability. Motivated by this, our work adopts a pretrain–finetune pipeline that extends physics-informed modeling to large-scale humanoid pretraining and safe finetuning.

## 3 PRELIMINARIES

**Problem Formulation.** RL is typically modeled as a Markov Decision Process (MDP) (Altman, 1999): $\mathcal{M} = (\mathcal{S}, \mathcal{A}, \mathbb{P}, R, \mu, \gamma)$, where $\mathcal{S}$ and $\mathcal{A}$ denote the state and action spaces, respectively. $\mathbb{P}(s'|s, a)$ and $R(r|s, a)$ represent the transition probability and reward function. $\mu(\cdot)$ denotes the initial state distribution, and $\gamma$ is the discount factor. A stochastic policy $\pi_\theta$ parameterized by $\theta$ specifies the action probability $\pi_\theta(a|s)$ for a given state $s$. In continuous control, the stochastic policy is typically Gaussian, $\pi_\theta(a|s) = \mathcal{N}(\mu_\theta(s), \Sigma_\theta(s))$, where the covariance can be either state-independent or state-dependent. During pretraining, domain randomization is employed to enhance robustness by perturbing the environment transition function $\mathbb{P}$ across episodes. At the same time, the stochastic policy is used to promote diverse exploration. For the deployment and finetuning stage, however, we adopt the deterministic policy induced by the mean action $\mu_\theta(s)$ to collect data and evaluate performance in environments.

**Model-based RL Problem.** For the finetuning stage, we seek a policy $\pi_\theta \in \Pi_\theta$ that maximizes the model-predicted return $J_\phi^R(\pi_\theta)$, defined as follows:

$$J_\phi^R(\pi_\theta) = \mathbb{E}\left[\sum_{t=0}^{\infty} \gamma^t r_{t+1} \,\middle|\, s_0 \sim \mu, \ a_t \sim \pi_\theta(\cdot|s_t), \ (s_{t+1}, r_{t+1}) \sim \mathbb{P}_\phi(\cdot \mid s_t, a_t)\right]. \tag{1}$$

Here $\mathbb{P}_\phi(\cdot|s_t, a_t)$ denotes a world model parameterized by $\phi$, which approximates the true environment dynamics. The initial state $s_0$ is sampled from the true initial distribution $\mu$, while subsequent states are generated by the learned model. In practice, a stochastic policy $\pi_\theta$ is used to interact with $\mathbb{P}_\phi$ so as to generate diverse imaginary trajectories, thereby improving exploration coverage for policy optimization. When the predictions of the world model are sufficiently accurate, maximizing the model-predicted return corresponds to maximizing the return in the real environment (Janner et al., 2019; Hafner et al., 2023).

**Physics-Informed World Model for Robots** Incorporating physics priors into the world model improves the accuracy of its predictions (Raissi et al., 2019). We adopt a physics-informed formulation of robot dynamics based on the Lagrangian equations of motion (Featherstone, 2008):

$$M(q_t)\ddot{q}_t + C(q_t, \dot{q}_t) + G(q_t) = B\tau_t + J^\top F_t^e + \tau_t^d, \tag{2}$$

Here, $q_t$ and $\dot{q}_t$ are the generalized coordinates and velocities at time $t$. The motor torque $\tau_t$ is applied to the joints via the matrix $B$. $F_t^e$ is the external contact force with Jacobian $J$, and $\tau_t^d$ denotes dissipative (parasitic) torques. The matrices $M$, $C$, and $G$ are the mass, Coriolis/centrifugal, and gravity terms. Because $B$, $M$, $C$, and $G$ depend only on the robot's shape and inertial parameters, they are known beforehand. Thus, the main model uncertainties are the contact term $J^\top F_t^e$ and the dissipative term $\tau_t^d$. To model these unknown contributions, Levy et al. (2024) introduce a residual network that approximates them. The network predicts the sum of the contact term and the dissipative term, as well as the uncertainty of the next-state prediction:

$$\tau_\phi(s_t, a_t) \approx \tau_t^e \equiv J^\top F_t^e + \tau_t^d \tag{3}$$

$$\sigma_\phi(s_t, a_t) = \log \sigma_t^2 \tag{4}$$

Here, $\tau_\phi$ and $\sigma_\phi$ are outputs from different heads of the neural network, with $\sigma_t^2$ representing the predicted variance of the next state. This hybrid physics-informed world model $\mathbb{P}_\phi$ combines the known rigid-body dynamics with learned residuals and can be rolled out to generate more precise trajectories for policy optimization.

## 4 Large-scale Pretraining and Efficient Fine-tuning

To enable a humanoid robot to learn quickly and with high sample efficiency in new environments, we first select the algorithm best suited for fine-tuning and then design the framework around it. Using a single algorithm across both stages aligns training objectives, enables continuous policy updates that mitigate forgetting, and provides a unified, easy-to-implement framework. While SSRL (Levy et al., 2024) successfully trained a quadruped robot from scratch using SAC and a physics-informed world model, its reliance on deterministic data collection leads to impractically slow run-times when scaled to humanoid robots. Combining this approach with PPO is a natural alternative, as PPO offers extensive parallel training infrastructure (Schwarke et al., 2025). However, our preliminary study (see Appendix A.1) highlights two advantages of SAC over PPO: its off-policy nature ensures stable convergence with limited samples, and its state-dependent stochastic actor promotes exploration in the world model, enhancing the diversity and utility of synthetic rollouts. Nevertheless, in our setting, none of these methods complete the target task within 48 hours of runtime without large-scale parallel pretraining, motivating the development of a full pretraining–finetuning pipeline using SAC as the backbone algorithm.

### 4.1 Large-scale Policy Pretraining

**Policy Optimization.** We adopt Soft Actor–Critic (SAC) (Haarnoja et al., 2018b) with an asymmetric actor–critic setup (Pinto et al., 2017): the actor $\pi_\theta(a_t|s_t)$ receives state $s_t$, while the critics $Q_{\psi_i}(s_t^p, a_t)$ are trained on privileged states $s_t^p$ that include additional information. our method uses

only the robot's proprioceptive state for both the actor and the critics. We normalize each feature of $s_t$ and $s_t^p$ using running mean and variance (Lee et al., 2024), ensuring balanced feature scales during training. We pretrain the SAC policy in MuJoCo Playground (Zakka et al., 2025) with thousands of vectorized environments on a single GPU. Our SAC implementation is written in JAX (Bradbury et al., 2018) with fixed tensor shapes, which allows efficient operation fusion and reuse of compiled kernels. As a result, large-batch updates (high UTD) incur no additional data transfer overhead. The same design seamlessly accelerates interaction with the world model during finetuning and supports multi-GPU scaling, improving wall-clock training time. Unlike FastTD3 (Seo et al., 2025) and PQL (Li et al., 2023), we do not use per-environment mixed Gaussian noise. Instead, we rely entirely on SAC's stochastic policy for exploration, i.e., $\pi_\theta(a \mid s) = \mathcal{N}(\mu_\theta(s), \Sigma_\theta(s))$, with state-dependent variance produced by the actor and temperature $\alpha$ controlling entropy. We employ the `Optuna` framework (Akiba et al., 2019) for systematic hyperparameter tuning. On the Booster T1 locomotion task, we conducted approximately ten hours of hyperparameter search on a single NVIDIA RTX 4090 GPU. The results show that, after tuning, the convergence time is reduced from about seven hours to only half an hour. The training objective and hyperparameter tuning process are detailed in Appendix B.1.

**Large-Batch Updates and High UTD.** Raising the update-to-data (UTD) ratio reuses experience more aggressively but can amplify value-estimation bias. In our `T1LowDimJoystick` setup—SAC trained in `MuJoCo Playground` with 1,024 parallel environments, large-batch updates (batch size 1,024), and a $10^6$-transition replay buffer—we observe that increasing UTD from 1 to 10 improves sample efficiency without any auxiliary stabilizers or architectural modifications. Beyond this range, gains diminish while wall-clock time increases predictably. We do not claim generality: these improvements appear tied to our massively parallel, domain-randomized regime, which likely suppresses early overfitting and curbs bias accumulation at moderate UTDs. By contrast, prior works typically couple high UTD with stabilizers such as periodic resets (D'Oro et al., 2023; Nikishin et al., 2022) or architectural regularization (ensemble/dropout critics, batch-normalized critics) (Chen et al., 2021; Hiraoka et al., 2022; Bhatt et al., 2024). We view these techniques as complementary and worth re-evaluating in large-scale parallel training for SAC to potentially push the usable UTD higher.

## 4.2 World Model Pretraining

**Offline Pretraining from Logged Transitions** Unlike MBPO or Dreamer, which update the policy and world model together online, we decouple them to improve wall-clock efficiency under massive parallelism. With thousands of parallel simulators, each step generates on the order of $\mathcal{O}(10^3)$ transitions, and training the world model online would slow the training. During SAC pretraining, we log all transitions to disk, saving records of the on-device replay buffer. After the policy converges, we train the world model entirely offline from these records. From each record, we construct $\mathcal{D} = (s_t^p, a_t, s_{t+\Delta t}^p)$. In our setting, the privileged state is $s_t^p = [q_t, \dot{q}_t, v_t^{\mathrm{b}}, \omega_t^{\mathrm{b}}, n_t, h_t]$, which includes joint angles and velocities, base linear and angular velocity in the body frame, base orientation, and body height.

**World Model and Loss.** We learn a residual predictor that injects external torques into a differentiable physics step. Residual predictor takes the concatenation of the privileged observation and action $[s_t^p, a_t]$ as input and outputs external uncertainties of torques and per-feature predictive uncertainty of the next-observation prediction:$(\tau_t^e, \log \sigma_t^2)$, as shown in Eq. (3). Given $[s_t^p, a_t, \tau_t^e]$, we predict the next privileged state using a fully differentiable pipeline. Concretely, we implement the world model based on Brax (Freeman et al., 2021) and perform the following steps: (a) map privileged state to Brax generalized coordinates and velocities, (b) convert actions to motor torques via a PD controller, (c) integrate the Lagrangian dynamics with semi-implicit Euler, and (d) reconstruct the next privileged state for learning and rollout. Using Brax's differentiable rigid-body primitives — $M(q)$, $C(q, \dot{q})$, and $G(q)$ — we avoid reimplementing the dynamics and keep the rollout loss pipeline end-to-end differentiable. See Appendix B.2 for details.

Compared to SSRL(Levy et al., 2024), we (i) correct the mapping from privileged state to Brax's generalized state $(q, \dot{q})$, resolving a discrepancy we identified in the publicly available SSRL implementation; and (ii) align each observation/state dimension between MuJoCo Playground and Brax (frames, quaternion form, normalization) to avoid training error;(iii) we include the base height $h_t$

in the privileged state. For humanoid tasks, omitting $h_t$ consistently caused unstable world model rollouts and non-convergent fine-tuning, whereas SSRL's quadruped configuration omitted height without issue. These adjustments improve training stability in humanoids setting.

After obtaining the next privileged state $\widehat{s}^p_{t+\Delta t}$ from the differentiable pipeline, we minimize the negative log-likelihood of a Gaussian predictive distribution. Let $B$ denote the batch size. For prediction $\widehat{s}^p_{t+\Delta t}$, target $s^p_{t+\Delta t}$, and elementwise log-variance $\log \sigma^2_t$, the loss is

$$\mathcal{L}_\phi = \frac{1}{B} \sum_{b=1}^{B} \left( \left(\widehat{s}^p_{b,t+\Delta t} - s^p_{b,t+\Delta t}\right)^2 \odot \exp\left(- \log \sigma^2_{b,t}\right) + \log \sigma^2_{b,t} \right), \tag{5}$$

where $\odot$ is elementwise multiplication. Gradients backpropagate through normalization, frame transforms, the PD controller, and the Euler Step, so the residual predictor is trained end-to-end from next-state errors. Minimizing this loss encourages the world model to predict small variance for seen states and large variance for unseen states. In practice, we exploit this behavior during rollouts by sampling next states from the predicted Gaussian $\widehat{s}^p_{t+\Delta t} \sim \mathcal{N}\left(\widehat{s}^p_{t+\Delta t}, \text{diag}(\sigma^2_t)\right)$.

### 4.3 POLICY AND WORLD MODEL FINETUNING

**Deployment & Data Collection.** We deploy the pretrained policy in the target environment (Brax (Freeman et al., 2021) in our case) and collect trajectories using deterministic actions (i.e., the action mean, without exploration noise). If the robot enters an unsafe state, the episode is reset. Each episode has a maximum length of $T_{\text{ep}} = 1000$. All transitions are stored in a replay buffer, which is subsequently used to fine-tune both the world model and the policy. After collecting $T_{\text{ep}}$ steps of data, we fine-tune the world model and policy before beginning the next iteration of data collection and training. More sophisticated pipelines, such as asynchronous data collection and learning (Luo et al., 2024), are left for future work to avoid additional complexity in this study.

**World-Model Fine-Tuning.** We fine-tune the pretrained world model on the collected buffer. For each epoch we independently shuffle data, roll physics inside the loss, and evaluate world model on a held-out testing data set for early stopping. We observe that training world model for multiple epochs on the same replay buffer, combined with auto-regressive training, enhances sample efficiency. The auto-regressive training objective is provided in the Appendix B.3. After World model training's done, we use the policy to explore in the world model to generate data that used to train the actor critic. Once world-model training is complete, we use the stochastic SAC actor to explore within the world model and generate trajectories.

**Exploration in Physics-informed World-Model.** *(a) Asymmetric Observation Generation.* Because data are collected with a deterministic policy (action mean only), the logged trajectories have limited diversity. To drive policy improvement during finetuning, we therefore explore inside the world model using a stochastic SAC policy. We keep the same asymmetric structure for actor-critic during finetuning as in pretraining: the actor consumes the proprioceptive state, while the critic uses the privileged state. Let $s^p_t \in \mathbb{R}^{d_p}$ denote the privileged state and $s_t \in \mathbb{R}^{d_s}$ the proprioceptive state, from which the action $a_t \sim \pi_\theta(\cdot|s_t)$ is sampled. The learned physics-informed world model $\mathbb{P}_\phi$ predicts the next privileged state, $\widehat{s}^p_{t+1} = \mathbb{P}_\phi(s^p_t, a_t)$ and the proprioceptive state is obtained by a fixed, simulator-consistent projection $\Pi : \mathbb{R}^{d_p} \to \mathbb{R}^{d_s}$ (same deterministic transforms used by the environment): $\widehat{s}_{t+1} = \Pi(\widehat{s}^p_{t+1})$. *(b) Safety Reset.* For safety and numerical robustness, we apply physics-informed terminal checks and terminate a world model rollout immediately upon detecting an abnormal state—specifically, when base height, world-frame linear/angular velocities, body roll/pitch, or joint positions/velocities violate prescribed bounds. Early termination stabilizes training and prevents NaNs over long horizons. We leave richer safeguards—e.g., self-collision checks derived from link poses—as future optimization. *(c) Physics-Consistent Reward.* Since $\widehat{s}^p_t$ provides all necessary quantities (joint states, link poses, base kinematics), rewards can be computed deterministically and remain fully consistent with the underlying physics in the world model rollout. An example is provided in the Appendix B.4. In contrast, using a learned scalar predictor $\hat{r}_\psi(o^p_t, a_t)$ (Hafner et al., 2023) is unstable in deterministic data collection settings. Even small state prediction errors accumulate over long horizons, degrading policy convergence. We sample a batch of privileged states from the replay buffer as initial conditions and roll out trajectories of horizon $H_{wm} = 20$ using the world model and the stochastic policy. These synthetic trajectories are added

to the replay buffer to train the actor-critic via SAC. The updated policy is then deployed in the environment to collect new data, thus completing one cycle of an iterative process that continues until convergence.

# 5 EXPERIMENTS

Our approach performs comparably to baselines in pretraining and, importantly, enables efficient fine-tuning for humanoids in new environments with limited data. We validate on two platforms—Booster T1 and Unitree G1—and report LIFT results averaged over 8 independent runs. All experiments are executed on a single NVIDIA 4090 GPU with 32 CPU cores in a cloud setting. Evaluation uses the average undiscounted return over $E$ episodes, $\hat{J}(\pi) = \frac{1}{E} \sum_{i=1}^{E} \sum_{t=0}^{T_{\text{ep}}} r_t$, computed without network updates. In practice, we set $E = 1024$ and $T_{\text{ep}} = 1000$. To characterize efficiency end-to-end, we compare (i) wall-clock time during pretraining (training speed) and (ii) environment steps required during fine-tuning (sample efficiency). These experiments are designed to answer three key questions: **(Q1)** Is the LIFT pretraining module comparable to established baselines? **(Q2)** Does this module better support subsequent fine-tuning than baselines? **(Q3)** Which components of our pipeline affect fine-tuning efficiency and stability?

**Baseline algorithms.** The baselines include: (1). **FastTD3** (Seo et al., 2025) (Model-free): An efficient variant of TD3 with parallel environments and mixed exploration noise. (2). **PPO** (Schulman et al., 2017) (Model-free): A widely used on-policy algorithm with clipped surrogate objective. We use baseline PPO in Mujoco playground. (3). **SAC** (Haarnoja et al., 2018b) (Model-free): An off-policy algorithm with stochastic actor and entropy regularization to encourage exploration. (4). **SSRL** (Levy et al., 2024) (Model-based): A physics-informed world model approach for training policies from scratch. (5). **MBPO** (Janner et al., 2019) (Model-based): Utilizes an ensemble of neural networks as the world model to improve sample efficiency of RL.

## 5.1 PRETRAINING EXPERIMENTS

To answer **Q1**, policy pretraining is performed in the MuJoCo Playground (Zakka et al., 2025) across six humanoid tasks, combining two terrain types (flat and rough) with three robot configurations: a low-dimensional Booster T1 (12-DoF, legs only), a full Booster T1 (23-DoF, including head, waist, arms, and legs), and the Unitree G1 (29-DoF, including waist, arms, and legs). Full details are provided in the Appendix C.2.

**Pretraining results.** As shown in the Appendix A.2, LIFT achieves comparable or higher evaluation returns than PPO and FastTD3 while stabilizes at its peak return faster on rough terrain environments. On flat terrain, it achieves comparable peak performance with similar wall-clock runtime. These curves indicate that the LIFT pretraining module is competitive in reward returns and convergence speed. Additionally, we also apply LIFT pretraining to whole-body tracking tasks for Unitree G1, providing preliminary evidence that the same framework extends naturally to non-locomotion skills. More details and discussion of these results are provided in Appendix A.8.

**Sim-to-real with LIFT Pretrained Policy.** To demonstrate the potential for zero-shot deployment, we deploy the pretrained policy for the low-dimensional Booster T1 directly on the physical robot. Zero-shot transfer to previously unseen surfaces (e.g., grass, uphill, downhill, mud, *etc.*) is shown in the Appendix A.6. This provides practical evidence that large-scale, parallel SAC pretraining can yield deployable humanoid controllers, thereby addressing **Q1**. And it also establishes a suitable starting point for subsequent fine-tuning.

## 5.2 FINETUNING EXPERIMENTS

**Sim-to-sim Finetuning.** We evaluate finetuning in Brax (Freeman et al., 2021) after large-scale policy pretraining in the MuJoCo Playground. The world model is also pretrained with data collected with pretrained policies. Because the two simulators differ in contact models and constraints, this sim-to-sim transfer provides a controlled but nontrivial test of adaptation. We transfer the pretrained policy to Brax and keep the reward design same as *Booster Gym* Wang et al. (2025) for both pretraining and finetuning. During pretraining, target linear velocities along the $x$-axis were uniformly sampled in $[-1, 1]$ m/s. For fine-tuning in Brax, we specify new forward-velocity targets and the

policy explores with deterministic action execution (action mean only; no stochastic sampling in the environment). We measure (i) whether policies converge within a fixed sample budget and (ii) velocity-tracking accuracy relative to the specified target. Implementation details are provided in the Appendix C.3.

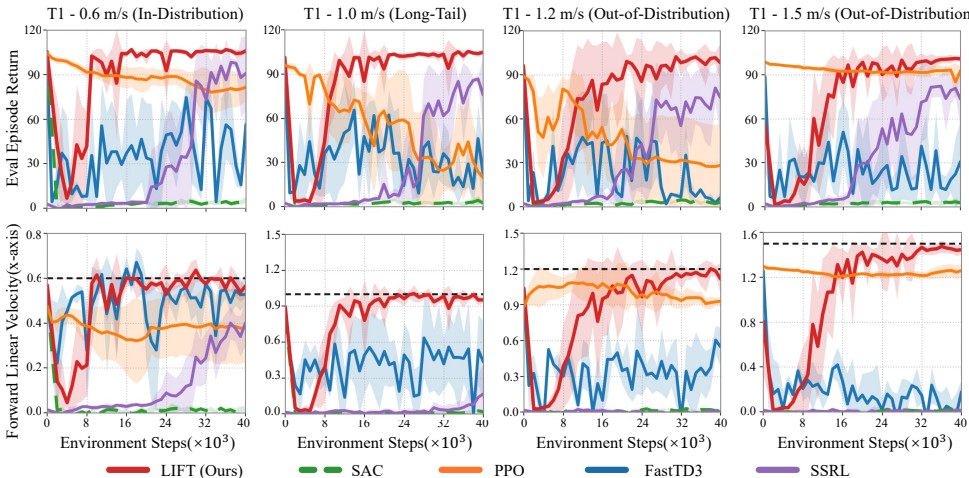

Figure 2: Results of finetuning Booster T1 robot with varying target speeds. The black dashed line represents the target velocity for each task. Results are averaged over 8 random seeds.

To address **Q2**, we design three scenarios and compare against PPO- and FastTD3-pretrained policies: **(i)In-Distribution**: targets within the pretraining range; **(ii) Long-Tail**: rare targets poorly represented during pretraining; **(iii) Out-of-Distribution**: targets outside $[-1, 1]$ m/s.

As shown in the Figure 2, LIFT consistently converges across all tasks, achieving stable walking that closely tracks the desired forward speed during the Booster T1 finetuning experiments. After finetuning, the policy demonstrates significantly reduced body oscillations and less deviation from the desired speed direction, with noticeable improvements in velocity. In contrast, SAC, trained without explicit exploration noise, quickly diverges and fails to recover, indicating rapid overfitting to the deterministically collected data. Although PPO's clipping mechanism stabilizes updates by keeping the new policy close to the previous one, its performance in our setting initially remains reasonable but then gradually degrades and ultimately collapses. This suggests that, under deterministic execution and limited data collection, PPO struggles to sustain stable policy improvement. FastTD3 exhibits strong oscillations and ultimately collapses without converging. SSRL shows signs of convergence at 0.6 m/s but fails to reach the target speed, and it does not converge at all on higher-speed tracking tasks (1.0–1.5 m/s). This highlights the difficulty of training humanoid locomotion from scratch and further motivates our pretrain–finetune design. These results demonstrate that LIFT enables robust adaptation under both in-distribution and out-of-distribution finetuning tasks, while standard baselines struggle with stability. During finetuning, policy control and data collection run at 50 Hz (one environment step = 0.02 s), so $4 \times 10^4$ steps correspond to approximately 800 s of on-robot interaction, highlighting the potential feasibility of applying LIFT on real humanoid robots which remains as future works. Additionally, we finetuned the Unitree G1 in the Brax environment, which also improved policy behavior and reward performance. The detailed results can be found in the Appendix A.7.

**Real-world Finetuning.** We evaluate LIFT on the Booster T1 humanoid by first pretraining a policy in the MuJoCo-Playground *T1LowDimJoystickFlatTerrain* task with most energy-related regularizations removed, keeping only an action-rate L2 penalty. This policy transfers well from MuJoCo to Brax but fails in zero-shot sim-to-real, providing a challenging starting point for real-world finetuning. Using this policy as initialization, our finetuning framework progressively improves initial unstable behavior: after collecting 80–590 s of data, the robot shows a more upright posture, smoother gait patterns, and more stable forward velocity (Fig. 3), demonstrating that LIFT can substantially strengthen a weak sim-to-real policy with only several minutes of real data. However, our current real-world implementation has two main *practical limitations*: (1) LIFT requires base-height es-

timation for world-model training and rewards, but Booster T1 does not provide this onboard, so we rely on a Vicon motion-capture system, which restricts the tracking area and requires human supervision. (2) We estimate base linear velocity by integrating IMU acceleration, which introduces drift and may limit policy tracking performance. (3) Each finetuning iteration is executed sequentially—up to 8 s of data collection at 50 Hz, world model update, then synthetic rollouts for policy updates—leading to multi-hour wall-clock time and frequent battery swaps despite using only minutes of real data. We view these as engineering rather than conceptual constraints, and expect that adopting an asynchronous pipeline similar to SERL (Luo et al., 2024), together with camera-based height and velocity estimation onboard, would make repeated real-world finetuning substantially more practical. More results are provided in Appendix A.3.

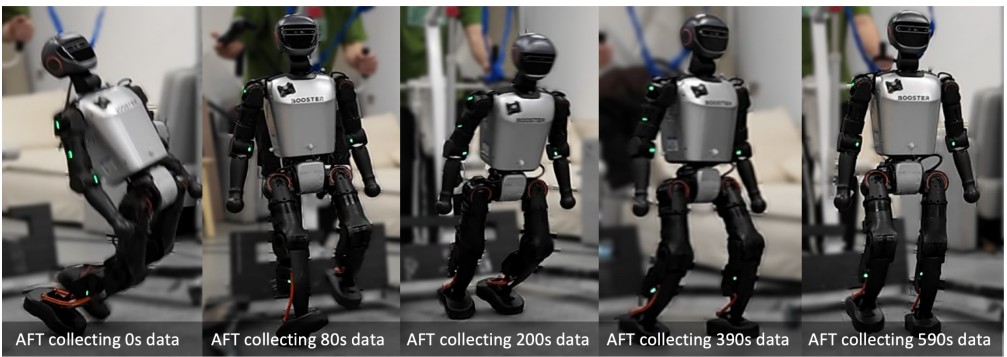

Figure 3: Real-world finetuning progression on the Booster T1 humanoid. A video demonstration is available on our project website.

## 5.3 ABLATION STUDY

To answer **Q3** and identify which components of LIFT affect fine-tuning efficiency and stability, ablations over three aspects are conducted: (i) *pretraining stages*—large-scale SAC pretraining and world-model (WM) pretraining; (ii) *world-model choice*—our physics-informed model versus an MBPO-style ensemble; and (iii) *key hyperparameters*—UTD ratio, replay-buffer size, batch size (pretraining), entropy coefficient $\alpha$ and autoregressive loss horizon (fine-tuning).

**Effect of Pretraining.** As shown in the Figure 4, we ablate WM and SAC pretraining. With both pretraining stages, LIFT converges within $4 \times 10^4$ environment steps and successfully tracks the target speed. Removing WM pretraining still allows the method to converge eventually, but noticeably slows down training, indicating that WM pretraining improves sample efficiency. Removing both pretraining stages reduces the LIFT to SSRL Levy et al. (2024), which mostly learns to stand in place with near-zero forward velocity. Thus, large-scale SAC pretraining is essential to avoid poor local minima, and WM pretraining further improves finetuning efficiency and stability.

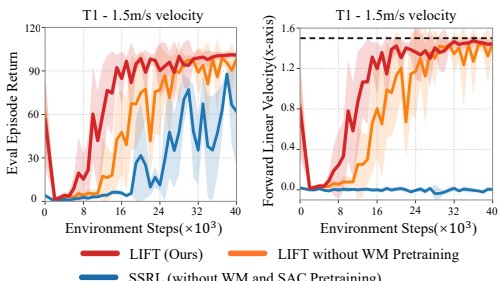

Figure 4: Ablation of the pretraining on Booster T1 (target forward speed = 1.5 m/s). Results are averaged over 8 random seeds.

**Physics-informed vs. Non-physics-informed World Models** We pretrain MBPO's ensemble world model (ensemble size = 5, elite size = 3) on the same dataset and finetune with identical hyperparameters to LIFT; the only difference is the choice of world model. MBPO fails to converge: episode return remain near zero (training curves as shown in Figure 5). On the test set its mean squared error (MSE) of world model is substantially worse than LIFT's. During model-based rollouts, a stochastic policy frequently produces actions that lie outside the distribution that has been seen in world-model training. These out-of-distribution actions induce physically implausible predictions (e.g., body height) for MBPO, which cause the critic loss to explode and inhibit policy improvement. This behavior likely stems from the purely neural network's limited ability to

generalize. By contrast, LIFT's physics-informed world model supplies strong inductive priors that improve generalization under limited data and produce stable, learnable rollouts, enabling successful finetuning.

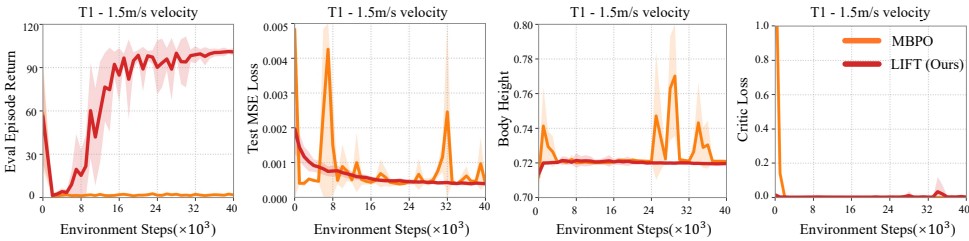

Figure 5: Ablation of Physics informed World Model on Booster T1 (target speed = 1.5 m/s). Results are averaged over 8 random seeds.

**Effect of Hyperparameters.** During pretraining, we find that the UTD ratio, buffer size, and batch size strongly influence convergence speed and performance. UTD = 1 converges very slowly, while increasing it to 5 speeds up learning. Larger buffer and batch sizes further accelerate convergence, though excessively large buffers increase GPU memory usage. Beyond certain thresholds, larger batch sizes or higher UTD provide little additional benefit. In finetuning, the entropy coefficient $\alpha$ and autoregressive horizon length are critical. Excessively large $\alpha$ promotes exploration into high-uncertainty regions of the world model, destabilizing learning. Using $\alpha$ from pretraining works reasonably, but smaller values achieve more stable convergence. A loss horizon length of 1 sometimes fails to reach the target velocity, while lengths of 2 or 4 consistently ensure stable learning, indicating that multi-step autoregressive prediction improves world-model training and policy stability. The training curves are provided in the Appendix A.4.

## 6 CONCLUSION AND DISCUSSION

In this work, we propose LIFT, a pretraining–finetuning framework for humanoid control that bridges large-scale simulation and data-efficient adaptation. By leveraging massively parallel environments with SAC, LIFT enables fast, robust pretraining and zero-shot sim-to-real transfer. The pretrained policy then guides model-based fine-tuning, while the same pretraining data bootstrap a physics-informed world model to improve sample efficiency. During fine-tuning, deterministic action execution is combined with stochastic exploration inside the world model, providing stable learning under limited data. Experimental results highlight a practical path toward continuous, efficient humanoid learning.

Despite these results, there still exist limitations in our current work that inspire several future directions: (1) *Safety management during real-world finetuning:* collecting deterministic trajectories in the real world still carries inherent risks due to model errors in the actor network. We enforce strict termination conditions, aligned with those in simulation, to halt episodes when key physical quantities exceed predefined thresholds. Human operators also remotely terminate any episode exhibiting unsafe behaviors (e.g., drifting toward obstacles). Upon termination, the policy is halted and the Booster T1 is switched to damping mode to restrict further motion. The robot is then guided back to the starting area using its default walking controller, followed by a standing controller to reset it to a consistent initial configuration. This procedure maintains stable initial states across episodes and mitigates distribution mismatch during data collection. Future work may incorporate more automated safety mechanisms—such as robot-assisted resetting (Hu et al., 2025), uncertainty-aware exploration (An et al., 2021; Yu et al., 2020), or recovery policies with safety switches (Thananjeyan et al., 2021; Smith et al., 2022a)—to further improve safety and reduce human intervention during real-world humanoid finetuning; (2) *high-dimensional external sensor and visual inputs:* LIFT currently operates exclusively on proprioceptive observations and does not incorporate camera or other high-dimensional sensory inputs, in contrast to vision-based frameworks such as DreamerV3 (Hafner et al., 2023). Scaling LIFT to tasks with vision-centric objectives—such as dexterous manipulation or object-centric control—will likely require latent world models capable of capturing dynamics beyond the robot's body state, which we view as a promising direction for future extensions.

## ACKNOWLEDGEMENT

This work was supported in part by the National Natural Science Foundation of China (No. 62403064, 62403063) and Shenzhen Science and Technology Program (No. ZDCY20250901094531003). We thank the engineering team from Booster Robotics for technical support.

**Reproducibility Statement.** We train all LIFT agents on a single NVIDIA RTX 4090 GPU. Our experiments use the MuJoCo Playground[1] and Brax[2] simulators. The source code and full results are available on our project website: `https://lift-humanoid.github.io/`. Our baseline implementations are adapted from the following open-source repositories: PPO[3], FastTD3[4], MBPO, and SSRL[5]. Detailed hyperparameters, additional experiments, and environment configurations are provided in Appendix A, B and C.

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

## A  ADDITIONAL EXPERIMENTS

### A.1  PRELIMINARY STUDY

We consider the practical case where the pretrained policy entirely fails in the deployment stage, finetuning must effectively learn a new policy from scratch, **with data collection constrained to deterministic policy**. To study this setting, we design a simple experiment using the BoosterT1 robot, which has 12 lower-body degrees of freedom. The reward is composed of three terms commonly used in humanoid locomotion: a swing-leg reference, body-height tracking, and linear velocity tracking. The policy is trained to achieve a target forward velocity of $0.2\,\text{m/s}$. The implementation details can be found in Appendix C.1.

As shown in Figure 6, we compare PPO and SAC within the SSRL (Levy et al., 2024) framework. The default setting uses SAC, which leverages a state-dependent stochastic actor to explore in the world model and improve policy performance. For comparison, we replace SAC with PPO (implemented in RSL-RL (Schwarke et al., 2025)) while keeping all other components fixed, tuning only the hyperparameters. We observe that the most critical factor for PPO performance is the initial action standard deviation (std), which strongly influences the distribution of states explored in the world model. A larger std makes training difficult to converge and tends to drive the policy into regions where the world model is poorly trained, while a smaller std leads to under-exploration and unstable convergence, with some seeds failing to learn at all. To mitigate this, we replace the PPO actor with the SAC actor that outputs both mean and std of the action distribution. This modification greatly improves stability, with all 8 seeds converging, but still requires two to three times more samples than SAC. Meanwhile, we run these experiments for 48 hours on a single Brax (Freeman et al., 2021) simulation. None of the methods converge to the target velocity within this time, highlighting the need for large-scale pretraining to reduce wall-clock time and avoid convergence to local minima.

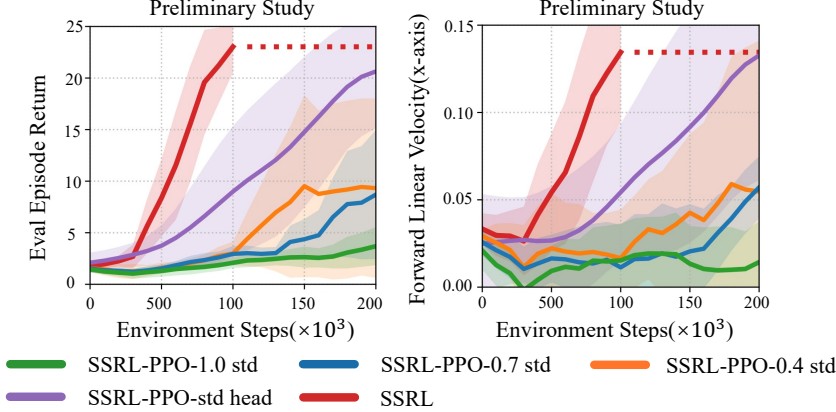

Figure 6: Comparison of PPO and SAC in a preliminary fine-tuning experiment with the BoosterT1 robot, run for 48 hours on a single Brax simulation. The left figure shows the evaluation episode return, and the right figure shows the body's linear velocity along the $x$-axis. Curves represent the average over 8 random seeds, with evaluation performed in 128 parallel environments. PPO performance is highly sensitive to the initial action standard deviation (std). Replacing the PPO actor with a SAC-style actor that outputs both the mean and standard deviation improves stability, although SAC still requires fewer samples to achieve the same performance.

## A.2 PRETRAINING EXPERIMENTS

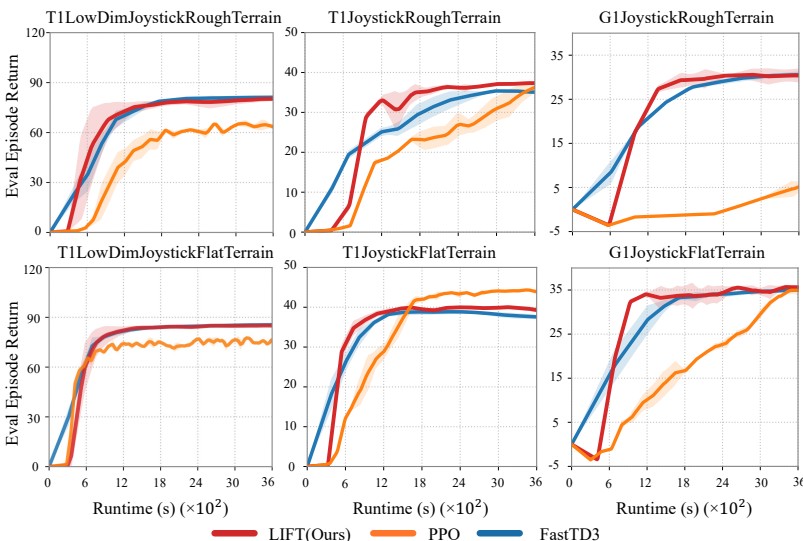

Figure 7: Pretraining performance comparison of LIFT (red), PPO (orange), and FastTD3 (blue) across six humanoid tasks (top row: rough terrain for the three robot configurations; bottom row: flat terrain for the three robot configurations). Results show the mean over 8 random seeds.

## A.3 FINETUNING EXPERIMENTS IN THE REAL-WORLD

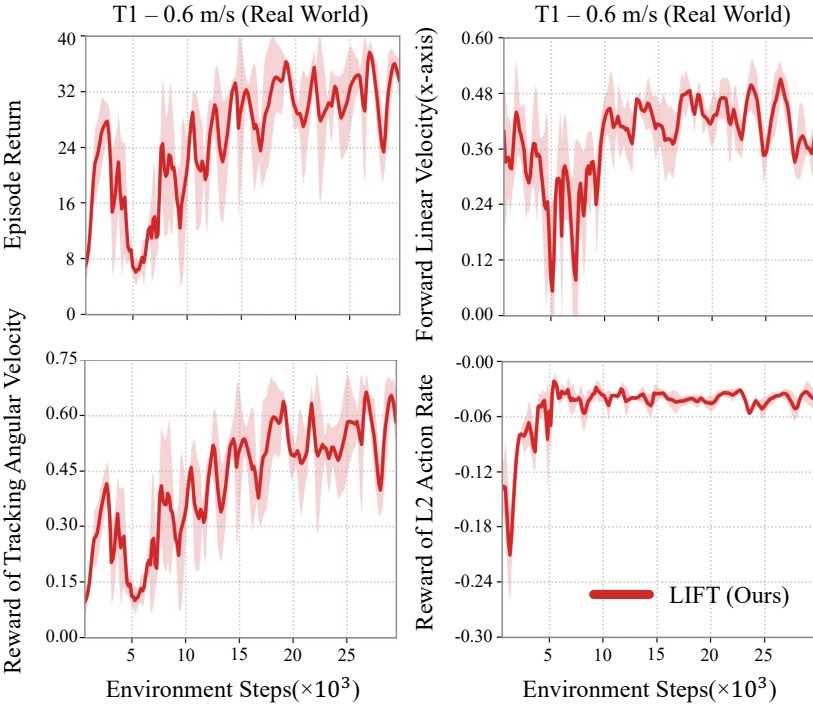

Figure 8: Finetuning performance of LIFT in the Real-World. Across 3 random seeds, LIFT consistently improves episode return, forward velocity tracking, and angular-velocity tracking, while reducing the action-rate penalty. Note that the velocity tracking error (target at 0.6 $m/s$) might come from the noisy base-velocity estimation from the onboard IMU accelerator.

## A.4 THE EFFECT OF HYPERPARAMETER

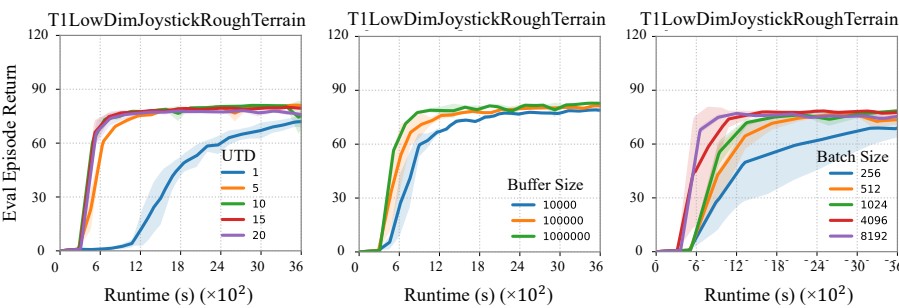

Figure 9: Effect of pretraining hyperparameters on Low dimensional Booster T1. From left to right: UTD ratio, buffer size, and batch size. UTD values compared: 1, 5, 10, 15, 20; buffer sizes: 10,000, 100,000, 1,000,000; batch sizes: 256, 512, 1,024, 4,096, 8,192. Larger UTD and batch sizes accelerate convergence; increasing buffer size improves learning speed but at the cost of much higher GPU memory usage.

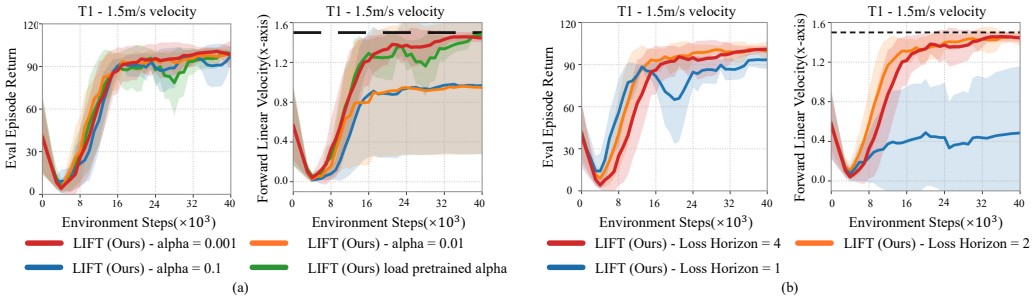

Figure 10: Impact of the entropy coefficient $\alpha$ and autoregressive loss horizon on finetuning performance (Booster T1, target speed = 1.5 m/s). Curves show evaluation episode return and body forward velocity. Larger $\alpha$ values degrade performance, leading to less stable convergence and lower final forward velocity. Using the pretraining $\alpha$ gives reasonable results, while smaller $\alpha$ yields more stable learning. Horizon = 1 sometimes fails to reach the target velocity, while horizons = 2 and = 4 ensure stable convergence, indicating that multi-step autoregressive prediction improves world-model training and stabilizes policy finetuning.

## A.5 PRETRAINING EXPERIMENTS

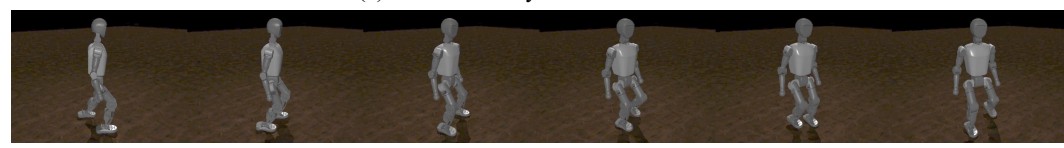

(a) T1LowDimJoystickFlatTerrain

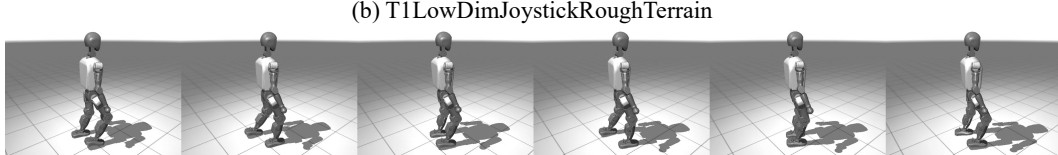

(b) T1LowDimJoystickRoughTerrain

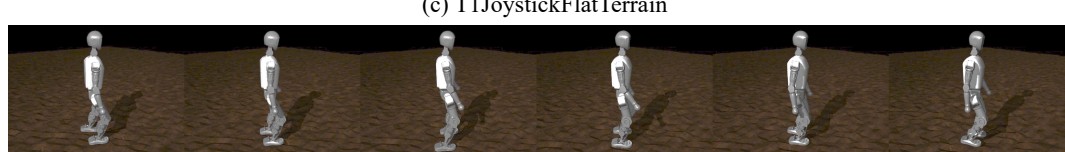

(c) T1JoystickFlatTerrain

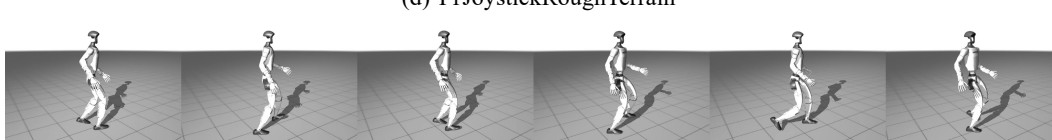

(d) T1JoystickRoughTerrain

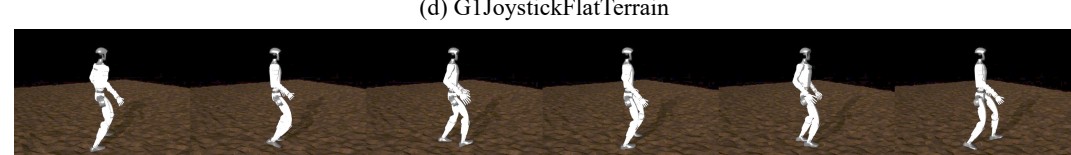

(d) G1JoystickFlatTerrain

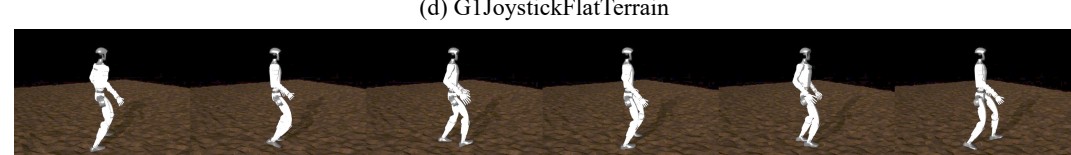

(f) G1JoystickRoughTerrain

Figure 11: Gaits of LIFT across different tasks in the MuJoCo Playground.

## A.6 REAL WORLD EXPERIMENTS

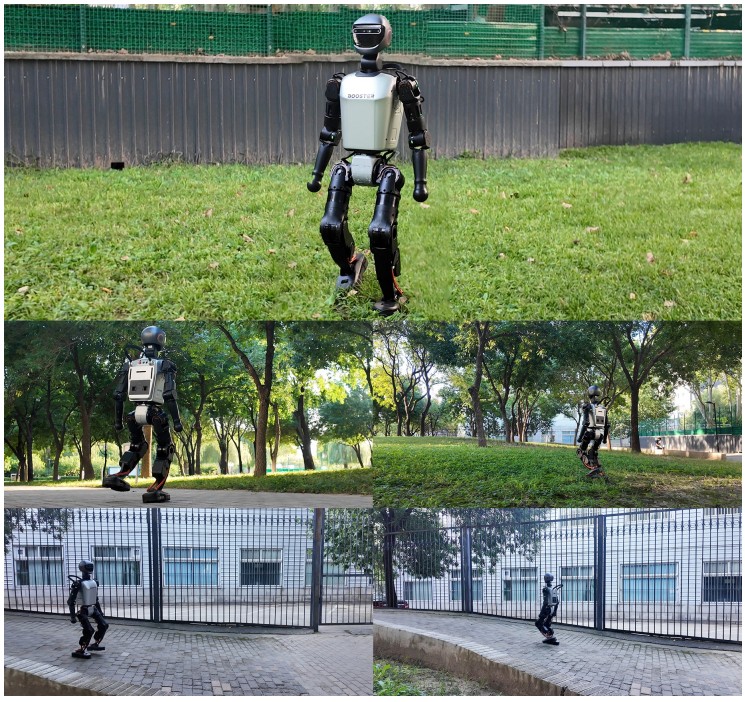

Figure 12: Sim-to-real reinforcement learning with LIFT.

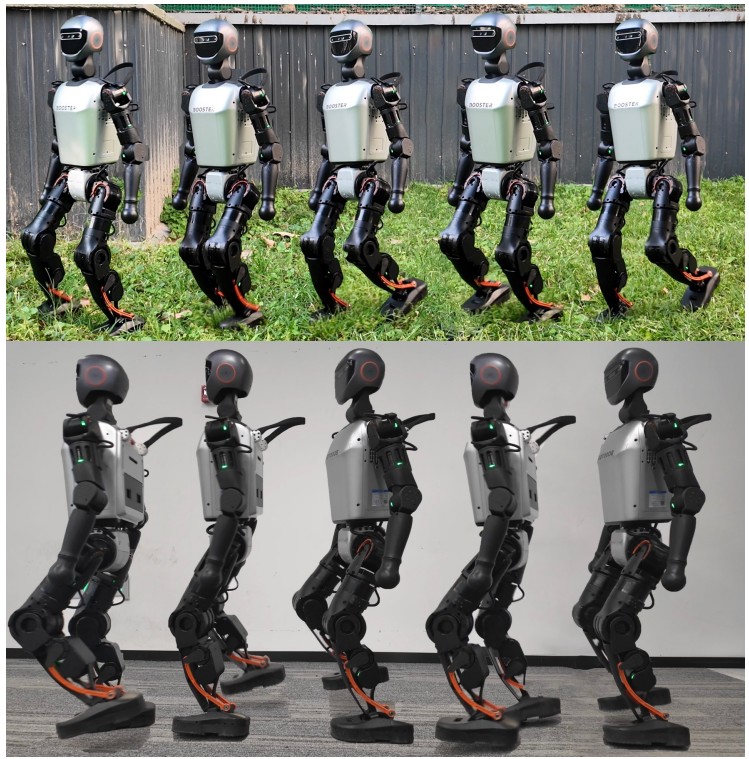

Figure 13: Sim-to-real gait comparison in indoor and outdoor environments.

## A.7 FINETUNING: G1 EXPERIMENTS

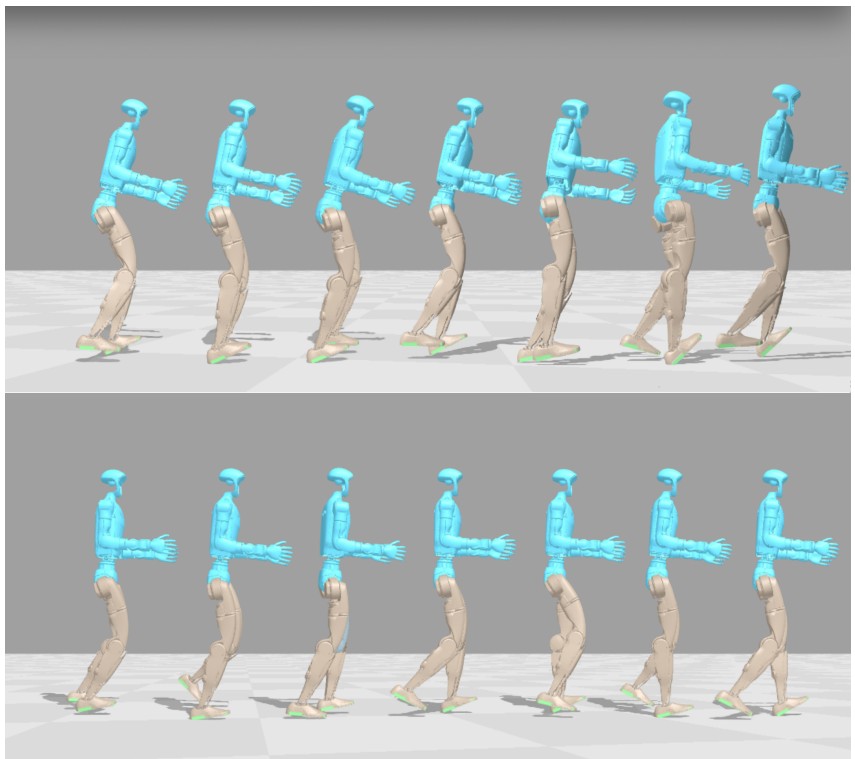

Figure 14: Sim-to-sim transfer and fine-tuning results for the Unitree G1 in Brax with a target velocity of 1.5 m/s. **Top:** Policy before fine-tuning, exhibiting instability and shuffling gait. **Bottom:** Policy after fine-tuning, demonstrating a stable, human-like walking gait with reduced torso pitch.

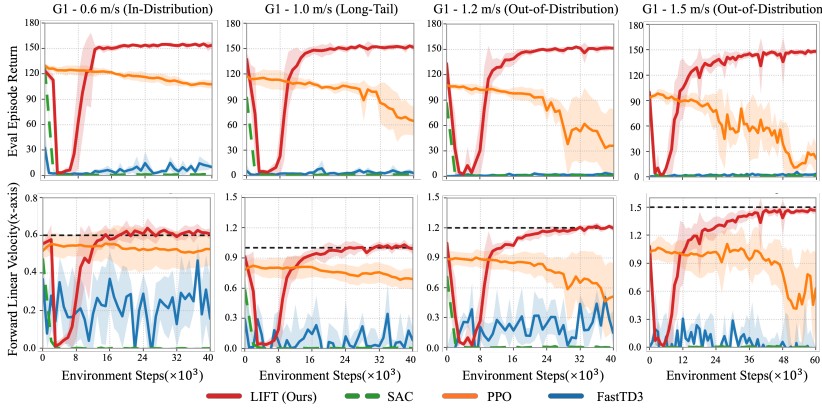

Figure 15: Finetuning results on the Unitree G1 in Brax. LIFT consistently improves policy behavior and reward performance. At a target speed of 0.6 m/s, body oscillations are significantly reduced, while at 1.5 m/s, the policy initially exhibits unstable motion in sim2sim but, after finetuning, successfully stands and walks.

### A.8 NON-LOCOMOTION TASKS

Our framework is built around learning a world model of the robot's state dynamics, so any task whose reward depends only on the robot proprioceptive state can, in principle, be handled directly. This includes different locomotion gaits, whole-body motion tracking, and balance under disturbances, since as long as the reward can be computed from the state, the policy can be finetuned entirely through world-model rollouts.

**Whole body motion tracking Tasks.** To better demonstraste this potential, we extended our pretraining pipeline from velocity-tracking locomotion to BeyondMimic-style whole-body tracking (Liao et al., 2025). We reimplemented the observation and reward structure of BeyondMimic in JAX within MuJoCo Playground and used the Unitree motion dataset (LAFAN1) to pretrain a whole-body tracking policy for the Unitree G1 humanoid. Video demonstrations are available on our project website. Due to the engineering effort required to reproduce consistent observation/reward definitions and contact handling across MuJoCo and Brax, and to integrate these components into our world-model finetuning pipeline, we were able to complete only the pretraining stage within the current timeframe. We therefore present these whole-body tracking results as preliminary evidence of LIFT's broader applicability. This limitation is practical rather than conceptual: once the corresponding Brax environment is finalized or real Unitree G1 hardware is available, the same stage-(iii) finetuning procedure can be directly applied.

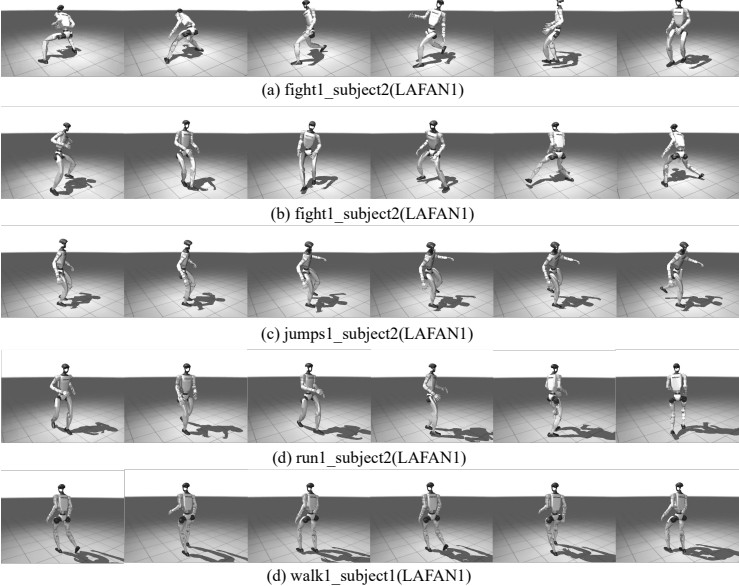

(a) fight1_subject2(LAFAN1)

(b) fight1_subject2(LAFAN1)

(c) jumps1_subject2(LAFAN1)

(d) run1_subject2(LAFAN1)

(d) walk1_subject1(LAFAN1)

Figure 16: Gaits of LIFT across different whole body motion tracking tasks in the MuJoCo Playground.

**Object-centric Tasks.** For tasks involving external objects (e.g., kicking a ball), the current framework would need to be extended to also model object dynamics, for example by augmenting the state with object pose and using simple physical priors such as conservation of momentum. This is feasible but requires additional engineering and environment design, so we explicitly leave it as future work. Likewise, obstacle avoidance and more complex navigation are more naturally handled at a higher level: once a reliable low-level tracking controller is available (as in our setup), a high-level policy can be trained to output velocity or pose targets, potentially with its own high-level world model as explored in hierarchical model-based RL (Hansen et al., 2024). Extending LIFT to hierarchical and object-centric tasks is a promising direction, but beyond the scope of this paper.

## B  TRAINING DETAILS

---

**Algorithm 1** The Finetuning phase of LIFT

---

**Require:** Pretrained policy $\pi$; pretrained world model $wm$; world-model replay buffer $D_{wm}$; batch size $B$; SAC replay buffer $D_{sac}$; maximum episode length $T_{ep}$; number of training iterations $num_{\text{train}}$; world-model rollout horizon $H_{wm}$.

1: **while** $\pi$ not converged **do** $s_0 = env.reset()$
2:    **for** $i \leftarrow 1$ to $T_{ep}$ **do**
3:        $a_t \sim \pi(\cdot|s_t)$
4:        $s_{t+1}, r_{t+1} = env.step(s_t, a_t)$
5:        $D_{wm} \leftarrow D_{wm} \bigcup \{s_t, a_t, s_{t+1}, r_{t+1}\}$
6:        $D_{sac} \leftarrow D_{sac} \bigcup \{s_t, a_t, s_{t+1}, r_{t+1}\}$
7:    **end for**
8:    Sample data from $D$ and finetune the world model using equation 19.
9:    **for** $g \leftarrow 1$ to $num_{\text{train}}$ **do** // $num_{\text{train}}$ *is linearly increased from 10 to 1000*
10:        Sample a batch of (Batch size $B$) initial state $s_0$ from the the $D_{wm}$.
11:        **for** $i \leftarrow 1$ to $H_{\text{wm}}$ **do** // $H_{\text{wm}}$ *is linearly increased from 1 to 20*
12:            $a_t \sim \pi(\cdot|s_t)$
13:            $s_{t+1}, r_{t+1} = wm.step(s_t, a_t)$ // *predict the next state and reward via world model.*
14:            $D_{sac} \leftarrow D_{sac} \bigcup \{s_t, a_t, s_{t+1}, r_{t+1}\}$
15:        **end for**
16:        Sample the transition from the $D_{sac}$, and update the SAC actor and critic.
17:    **end for**
18: **end while**
19: **return** $\pi$

---

### B.1  PRETRAINING DETAILS OF SAC

**Training Objective**    We pretrain the SAC policy in MuJoCo Playground. The critics minimize the entropy-regularized Bellman residual:

$$\mathbb{E}_{(s_t^p, a_t, r_{t+1}, s_{t+1}^p) \sim \mathcal{D}, a_{t+1} \sim \pi_\theta(\cdot|s_{t+1})} \left[ Q_\psi(s_t^p, a_t) - (r_{t+1} + \gamma(\min_{i=1,2} Q_{\bar{\psi}_i}(s_{t+1}^p, a_{t+1}) - \alpha \log \pi_\theta(a_{t+1}|s_{t+1})))^2 \right],$$

(6)

where $Q_{\bar{\psi}_i}$ denotes the target critics. Transitions are sampled from the replay buffer $\mathcal{D}$. The actor is updated by maximizing the entropy-augmented Q-value:

$$\mathbb{E}_{s_t, s_t^p \sim \mathcal{D}, a_t \sim \pi_\theta(\cdot|s_t)}[\alpha \log \pi_\theta(a_t|s_t) - \min_{i=1,2} Q_{\psi_i}(s_t^p, a_t)].$$

(7)

The entropy coefficient $\alpha$ is adjusted to match a target entropy $\bar{\mathcal{H}}$ via

$$\mathbb{E}_{s_t \sim \mathcal{D}, a_t \sim \pi_\theta(\cdot|s)}[-\alpha(\log \pi_\theta(a_t|s_t) + \bar{\mathcal{H}})].$$

(8)

Finally, the target critics are updated by Polyak averaging: $\bar{\psi} \leftarrow \tau \psi + (1 - \tau)\bar{\psi}$, where $\tau \in (0, 1)$ is the update rate.

**Hyperparameter-Tuning**    We employ the `Optuna` framework (Akiba et al., 2019) for systematic hyperparameter tuning, leveraging the `CmaEsSampler` for parameter exploration in combination with a `PatientPruner` (built on top of a `SuccessiveHalvingPruner`) to automatically terminate trials with poor early performance. The tuned hyperparameters include the number of gradient updates per step (UTD), replay buffer size, entropy coefficient $\alpha$, discount factor $\gamma$, learning rates of the actor, critic, and temperature parameter $\alpha$, batch size, reward scaling, number of parallel environments, and Polyak coefficient $\tau$. The optimization objective is the episode reward obtained during policy evaluation. On the `T1LowDimJoystick` tasks, we conducted approximately ten hours of hyperparameter search on a single NVIDIA RTX 4090 GPU. The results indicate that tuning reduces the convergence time from about seven hours to only half an hour. On the `G1Joystick` and `T1Joystick` tasks, we carried out a similar ten-hour search using eight NVIDIA RTX 4090 GPUs, which led to improved reward performance and more robust convergence. Table 1 lists the tuned hyperparameters.

Table 1: LIFT Hyperparameter Configurations for MuJoCo Playground Environments

| Hyperparameter | T1LowDimJoystick* | G1/T1 Joystick* | T1JoystickFlatTerrain |
|---|---|---|---|
| Parallel environments | 1,000 | 4,096 | 4,096 |
| Batch size | 1,024 | 16,384 | 16,384 |
| Gradient updates/step | 9 | 19 | 16 |
| Discount factor ($\gamma$) | 0.987 | 0.982 | 0.982 |
| Reward scaling | 1.0 | 16.0 | 32.0 |
| Soft update ($\tau$) | 0.024 | 0.002 | 0.007 |
| Actor learning rate | 1.03e-4 | 1.70e-4 | 2.05e-4 |
| Critic learning rate | 1.00e-4 | 1.88e-4 | 1.37e-4 |
| Alpha learning rate | 9.97e-3 | 6.44e-4 | 6.41e-4 |
| Target entropy coefficient | 0.50 | 0.40 | 0.17 |
| Initial log $\alpha$ | -3.35 | -3.00 | -6.05 |
| Policy hidden layers | (512, 256, 128) | (512, 256, 128) | (512, 256, 128) |
| Q-network hidden layers | (1024, 512, 256) | (1024, 512, 256) | (1024, 512, 256) |
| Activation | swish | swish | swish |
| Max replay size ($\times 10^6$) | 1.0 | 1.0 | 1.0 |
| Min replay size | 8,192 | 8,192 | 8,192 |
| Observation normalization | True | True | True |
| Action repeat | 1 | 1 | 1 |

## B.2 TRAINING DETAILS OF WORLD MODEL

*(a) From Privileged State to Brax Generalized State.* We build the *generalized coordinates/velocities* of Brax using the privileged state and denote them simply by $(q_t, \dot{q}_t)$ for the rest of the section:

$$q_t = [p_t, \; n_t, \; q_{j,t}], \quad p_t = (0, 0, h_t); \tag{9}$$

$$\dot{q}_t = [v_t^{\mathrm{w}}, \; \omega_t^{\mathrm{b}}, \; \dot{q}_{j,t}], \quad v_t^{\mathrm{w}} = R(n_t)\, v_t^{\mathrm{b}}, \tag{10}$$

where $R(n) \in \mathrm{SO}(3)$ be the rotation matrix from the base quaternion $n$ (body→world), with $R(q)^\top$ its inverse. Brax exposes differentiable rigid-body dynamics primitives—mass matrix $M(q)$, Coriolis/centrifugal terms $C(q, \dot{q})$, and gravity $G(q)$—thereby avoiding any reimplementation of dynamics and keeping the entire rollout–loss pipeline end-to-end differentiable.

*(b) From Privileged State and Action to Torque (repeat $N$ substeps).* The policy action is converted to motor torques $\tau_t^{\mathrm{m}}$ via a PD controller: $\tau_t^{\mathrm{m}} = K_p\, (a_t + q_t^{\mathrm{default}} - q_{j,t}) + K_d\, (0 - \dot{q}_{j,t})$, where $K_p$ and $K_d$ are stiffness and damping; $q_t^{\mathrm{default}}$ is default/standing joint position. The external torque $\tau_t^e$ come from the network $\tau_\phi(s_t^p, a_t)$. We assume the control rate $f_{\mathrm{ctrl}}$ is lower than the simulator/collection rate $f_{\mathrm{sim}}$. Hence each control interval contains $N = \frac{f_{\mathrm{sim}}}{f_{\mathrm{ctrl}}}$ simulator substeps of size $\Delta t_{\mathrm{sub}}$, so that the control interval length is $\Delta t = N\, \Delta t_{\mathrm{sub}}$.

*(c) One Semi-implicit Euler Step (repeat $N$ substeps).* For substep $k = 0, \ldots, N - 1$ with step size $\Delta t_{\mathrm{sub}}$:

$$\ddot{q}_t = M^{-1}(q_t)[\tau_t^m + \tau_t^e - C(q_t, \dot{q}_t) - G(q_t)], \tag{11}$$

$$\dot{q}_{t+\Delta t_{\mathrm{sub}}} = \dot{q}_t + \Delta t_{\mathrm{sub}}\, \ddot{q}_t, \tag{12}$$

$$q_{j,t+\Delta t_{\mathrm{sub}}} = q_{j,t} + \dot{q}_{t+\Delta t_{\mathrm{sub}}} \tag{13}$$

$$p_{t+\Delta t_{\mathrm{sub}}} = p_t + \Delta t_{\mathrm{sub}}\, v_{t+\Delta t_{\mathrm{sub}}}^{\mathrm{w}}, \tag{14}$$

$$n_{t+\Delta t_{\mathrm{sub}}} = \mathrm{normalize}\Big(n_t \otimes \mathrm{quat}\big(\widehat{\omega_{t+\Delta t_{\mathrm{sub}}}^{\mathrm{b}}}, \|\omega_{t+\Delta t_{\mathrm{sub}}}^{\mathrm{b}}\| \, \Delta t_{\mathrm{sub}}\big)\Big). \tag{15}$$

Here, $\widehat{\omega} = \omega/\|\omega\|$ is the unit rotation axis; $\mathrm{quat}(\widehat{\omega}, \theta) = [\cos(\frac{\theta}{2}), \widehat{\omega} \sin(\frac{\theta}{2})]$ is the axis–angle increment as a unit quaternion; $\otimes$ denotes quaternion multiplication; $\mathrm{normalize}(\cdot)$ re-unitizes the quaternion to avoid numerical drift. After $N$ substeps, we obtain:

$$q_{t+\Delta t} = [p_{t+\Delta t}, \; n_{t+\Delta t}, \; q_{j,t+\Delta t}], \tag{16}$$

$$\dot{q}_{t+\Delta t} = [v_{t+\Delta t}^{\mathrm{w}}, \; \omega_{t+\Delta t}^{\mathrm{b}}, \; \dot{q}_{j,t+\Delta t}]. \tag{17}$$

*(d) From $(q_{t+\Delta t}, \dot{q}_{t+\Delta t})$ to the Next Privileged State.:*

$$\widehat{s}^p_{t+\Delta t} : \begin{cases} \text{Joints Position/Velocity: } (q_{j,t+\Delta t}, \dot{q}_{j,t+\Delta t}). \\ \text{Base Linear Velocity: } v^{\mathrm{b}}_{t+\Delta t} = R(n_{t+\Delta t})^\top v^{\mathrm{w}}_{t+\Delta t}. \\ \text{Base Angular Velocity: } \omega^{\mathrm{b}}_{t+\Delta t}. \\ \text{Quaternion: } n_{t+\Delta t}. \\ \text{Body Height: } h_{t+\Delta t} = p_{t+\Delta t, z}. \end{cases} \tag{18}$$

### B.3 TRAINING DETAILS OF WORLD MODEL IN FINETUNING STAGE

Concretely, we sample length-$H{+}1$ sequences $\mathcal{D}_H = \left\{ (s^p_t, a_t, s^p_{t+1}, a_{t+1}, \ldots, s^p_{t+H}) \right\}^B_{b=1}$, where $B$ is the mini-batch size and $H$ is the loss horizon (e.g., $H{=}4$). Let $\widehat{s}^p_{b,t+1}$ be the model's one-step prediction (obtained by differentiable physics given $(s^p_{b,t}, a_{b,t})$), $s^p_{b,t+1}$ the target, and $\log \sigma^2_{b,t}$ the model's elementwise log-variance. We minimize an auto-regressive diagonal-Gaussian negative log-likelihood over an $H$-step unroll:

$$\mathcal{L}_\phi = \frac{1}{B\,H} \sum_{b=1}^B \sum_{t=0}^{H-1} \left[ \left( \widehat{s}^p_{b,t+1} - s^p_{b,t+1} \right)^2 \odot \exp\left( -\log \sigma^2_{b,t} \right) + \log \sigma^2_{b,t} \right], \tag{19}$$

where $\odot$ denotes elementwise multiplication, $\widehat{s}^p_{b,t+1}$ is the model's next-step prediction given $(s^p_{b,t}, a_{b,t})$, $s^p_{b,t+1}$ is the target, and $\log \sigma^2_{b,t}$ is the elementwise log-variance. Gradients backpropagate through the entire $H$-step unroll.

### B.4 TRAINING DETAILS OF POLICY IN FINETUNING STAGE

A key advantage of our physics-informed design is that $\widehat{s}^p_t$ exposes kinematic and dynamic signals that enable *exact*, simulator-consistent reward computation without training a reward model. We thus define a per-step reward:

$$r_{t+1} = \sum_k w_k\, r_k\left( \widehat{s}^p_t, a_t, \widehat{s}^p_{t+1} \right), \tag{20}$$

with each $r_k$ computed analytically. For example, foot orientation stability follows directly from each foot's roll angle and is penalized by

$$r_{\text{feet-roll}} = -\sum_{f \in \{\text{Left foot}, \text{Right foot}\}} \left( \varphi^f \right)^2, \tag{21}$$

where $\varphi^f$ is the roll of foot $f$ extracted from the link quaternion in $\widehat{s}^p_t$. Table 1 lists the LIFT hyperparameters in finetuning.

Table 2: Finetune Hyperparameter of LIFT

| Hyperparameter | Actor-Critic | World Model |
|---|---|---|
| Policy hidden layers | [512, 256, 128] | — |
| Q-network hidden layers | [1024, 512, 256] | — |
| World Model hidden size | — | [400,400,400,400] |
| Activation | swish | swish |
| Learning rate | 2e-4 | 1e-3 |
| Discount factor ($\gamma$) | 0.99 | — |
| Batch size | 256 | 200 |
| Soft update ($\tau$) | 0.001 | — |
| Initial log $\alpha$ | -7.13 | — |
| Gradient updates/step | 20 | — |
| Real data ratio | 0.06 | — |
| Init exploration steps | 1,000 | — |
| Replay buffer size | 400k | 60k |
| Reward scaling | 1.0 | — |
| Model trains/epoch | 1 | — |
| Env steps/training | 1,000 | — |
| Horizon of exploration in world model | 1→20 (epochs 0→10) | — |
| Gradient step per update | 10→1000 (epochs 0→4) | — |
| Convergence criteria | — | 0.01 (6 epochs) |
| Loss horizon | — | 4 |

## C  Environment Setup

### C.1  Preliminary Study - Booster Environment

**Task summary.** The agent controls a 12-DoF T1 humanoid robot with simplified lower-body kinematics. At each step, the agent outputs a 12D continuous action that perturbs motor target positions relative to a default pose.

**State (39D).** The state consists of torso quaternion (4), joint angles (12), base linear velocity in body frame (3), base angular velocity in body frame (3), joint velocities (12), gait phase represented by cosine/sine (2), gait progression (1), gait frequency (1), and base height (1).

**Privileged state (39D).** The privileged state uses the same 39-dimensional observation space as the state.

**Action Space.** The action space is a continuous 12-dimensional vector, where policy outputs are constrained to the range $[-1, 1]$. Motor targets are computed using PD control:

$$u = K_p(a \cdot a_s + q_{\text{default}} - q) + K_d(\dot{q}_{\text{des}} - \dot{q}) \tag{22}$$

where $a_s = 0.25$ is the action scale factor. The proportional and derivative gains are defined as:

$$K_p = [200, 200, 200, 200, 50, 50, 200, 200, 200, 200, 50, 50] \quad K_d = [5, 5, 5, 5, 1, 1, 5, 5, 5, 5, 1, 1]$$

The gains are symmetric for both legs, with higher gains for hip joints (200 for $K_p$, 5 for $K_d$) and lower gains for knee joints (50 for $K_p$, 1 for $K_d$).

**Reward Function Design.** Uses a multi-objective weighted reward function:

$$r_t = \Delta t \sum_i s_i r_i(s_t, a_t)$$

**Control Architecture.** Uses a two-level control strategy:

1. **High-level Policy:** Outputs joint position increments, control frequency 100 Hz

Table 3: Reward Function Components and Weights

| Reward Term | Description | Default Weight |
|---|---|---|
| base_height | Base height reward | 0.2 |
| tracking_lin_vel | Linear velocity tracking reward | 1.2 |
| ref | Reference trajectory tracking reward | 3.0 |

**Gait Generation.** Phase-based gait controller:

$$\phi = \mathrm{mod}(t \cdot f_g, 1.0)$$
$$\text{Left Leg Swing Phase} = |\phi - 0.25| < 0.5 \cdot T_{swing}$$
$$\text{Right Leg Swing Phase} = |\phi - 0.75| < 0.5 \cdot T_{swing}$$

where $T_{swing} = 0.2$s is the swing phase duration.

**Termination Conditions.** Episode terminates under the following conditions:

- Base height $h \notin [0.3, 0.8]$ m
- Base linear velocity $> 10.0$ m/s or angular velocity $> 10.0$ rad/s
- Torso roll angle $|\phi| > \pi/4$ or pitch angle $|\theta| > \pi/4$
- Joint position or velocity exceeds mechanical limits

**Experimental Setup.** In the Preliminary Study (Section A.1):

- Target forward velocity: 0.2 m/s
- Main reward terms: swing leg reference, base height tracking, linear velocity tracking

## C.2 PRETRAINING ENVIRONMENTS

### C.2.1 T1LOWDIMJOYSTICK

(`T1LowDimJoystick` as an example to illustrate the pretraining environment setup, `T1Joystick` and `G1Joystick` are omitted for brevity.)

**Task summary.** The agent controls a 12-DoF T1 humanoid robot with simplified lower-body kinematics. At each step, the agent outputs a 12D continuous action that perturbs motor target positions relative to a default pose. The environment returns noisy states and a scalar reward. Episodes terminate upon falls or numerical errors (NaNs).

**State (47D).** The state consists of gravity vector in the body frame (3), gyroscope readings (3), joystick command $(v_x, v_y, \omega)$ (3), gait phase represented by cosine/sine (2), joint angles relative to the default pose (12), joint velocities scaled by 0.1 (12), and previous action (12).

**Privileged state (110D).** The privileged state includes all elements of the 47D state, together with additional raw sensor signals (gyroscope, accelerometer, gravity vector, linear velocity, global angular velocity), joint differences, root height, actuator forces, contact booleans, foot velocities, and foot air time.

**Action space.** The action is a continuous 12-D vector; policy outputs lie in $[-1, 1]$. Motor targets are computed using a PD control law:

$$u = K_p(a + q_{\text{default}} - q) + K_d(0 - \dot{q}) \tag{23}$$

where $u$ is the output motor torque, $K_p$ and $K_d$ are the proportional and derivative gains, respectively, $a$ is the action output from the policy $q_{\text{default}}$ is a nominal joint position, $q$ is the current joint position, and $\dot{q}$ is the current joint velocity.

**Reward** Instantaneous reward: $r_t = \Delta t \sum_i s_i r_i(s_t^p, a_t)$, where $s_i$ are term weights and $\Delta t = 0.02$s.

Table 4: Default reward terms (Note: $U[a, b]$ denotes uniform distribution over $[a, b]$)

| Term | Description | Scale |
|---|---|---|
| survival | Survival bonus per step. | 0.25 |
| tracking_lin_vel_x | Track commanded $v_x$ velocity. | 1.0 |
| tracking_lin_vel_y | Track commanded $v_y$ velocity. | 1.0 |
| tracking_ang_vel | Track commanded yaw rate. | 2.0 |
| feet_swing | Reward proper foot swing phase. | 3.0 |
| base_height | Penalize deviation from target height (0.68m). | -20.0 |
| orientation | Penalize torso tilt from vertical. | -10.0 |
| torques | Penalize large torques. | -1.0e-4 |
| torque_tiredness | Penalize torque near limits. | -5.0e-3 |
| power | Penalize positive mechanical power. | -1.0e-3 |
| lin_vel_z | Penalize vertical base velocity. | -2.0 |
| ang_vel_xy | Penalize torso roll/pitch rates. | -0.2 |
| dof_vel | Penalize joint velocity. | -1.0e-4 |
| dof_acc | Penalize joint acceleration. | -1.0e-7 |
| root_acc | Penalize root link acceleration. | -1.0e-4 |
| action_rate | Penalize rapid action changes. | -0.5 |
| dof_pos_limits | Penalize joint limit violations. | -1.0 |
| collision | Penalize self-collisions. | -10.0 |
| feet_slip | Penalize slipping contacts. | -0.1 |
| feet_roll | Penalize foot roll angles. | -1.0 |
| feet_yaw_diff | Penalize difference in foot yaw angles. | -1.0 |
| feet_yaw_mean | Penalize deviation from base yaw. | -1.0 |
| feet_distance | Penalize feet being too close. | -10.0 |

**Implementation details**

- Control period: `ctrl_dt` = 0.02 s; simulation step: `sim_dt` = 0.002 s.
- External pushes enabled with interval 5.0-10.0s and magnitude 0.1-1.0N.
- Enhanced tracking rewards with separate x/y linear velocity tracking.
- Sophisticated foot kinematics penalties including roll, yaw, and swing phase rewards.

Table 5: Domain randomization parameters (Note: $U[a, b]$ denotes uniform distribution over $[a, b]$, $\pm U(c)$ denotes uniform distribution over $[-c, c]$)

| Parameter | Range | Description |
|---|---|---|
| Floor friction | $U[0.2, 0.6]$ | Ground friction coefficient |
| Joint friction loss | $\times U[0.9, 1.1]$ | Joint friction scaling |
| Joint armature | $\times U[1.0, 1.05]$ | Joint armature scaling |
| Link masses | $\times U[0.98, 1.02]$ | Body mass scaling |
| Torso mass | $+U[-1.0, 1.0]$ kg | Additional torso mass |
| Initial joint positions | $\pm U[0.05]$ rad | Default pose randomization |
| Joint stiffness | $\times U[0.7, 1.3]$ | Actuator gain scaling |
| Joint damping | $\times U[0.7, 1.3]$ | Damping coefficient scaling |
| Ankle damping | $\times U[0.5, 2.0]$ | Higher range for ankle joints |

## C.3 Finetuning Environments

### C.3.1 T1LowDimJoystick (Finetune)

(`T1LowDimJoystick` as an example to illustrate the finetuning environment setup, `G1LowDimJoystick` is omitted for brevity.)

**Task summary.** This is a simplified version of the T1LowDimJoystick environment designed specifically for sim-to-sim transfer from MuJoCo Playground to Brax and finetuning in Brax. The environment ensures that policies trained in MuJoCo Playground can be successfully transferred to Brax with similar reward scales and performance characteristics. The reward settings are adopted from Booster Gym (Wang et al., 2025).

**Key modifications for finetuning:**

- **Normalized observations** using predefined limits for consistent scaling across simulators
- **Simplified reward structure** with many terms zeroed out to focus on essential behaviors
- **Continuous gait phase** representation using gait process and frequency

**State (47D, normalized).** The normalized state consists of gravity vector (3), gyroscope readings (3), joystick command $(v_x, v_y, \omega)$ (3), gait phase cosine/sine (2), joint angles (12), joint velocities (12), and previous action (12). All observations are normalized to $[-1, 1]$ using predefined limits.

**Privileged state (87D, normalized).** Includes base state plus raw gyroscope, gravity vector, orientation quaternion (4), base velocity, joint positions/velocities, root height, gait process, and gait frequency.

**Observation normalization:** Observations are normalized using:

$$\text{obs\_normalized} = \frac{2 \times (\text{obs} - \text{min})}{\text{max} - \text{min}} - 1$$

with predefined limits for each observation dimension.

**Action space.** Motor targets are computed using a PD control law:

$$u = K_p(a + q_{\text{default}} - q) + K_d(0 - \dot{q}) \tag{24}$$

where $u$ is the output motor torque, $K_p$ and $K_d$ are the proportional and derivative gains, respectively, $a$ is the action output from the policy $q_{\text{default}}$ is a nominal joint position, $q$ is the current joint position, and $\dot{q}$ is the current joint velocity.

Table 6: Finetune reward terms (Note: many terms disabled for sim-to-sim transfer)

| Term | Description | Scale |
|------|-------------|-------|
| survival | Survival bonus per step. | 0.25 |
| tracking_lin_vel_x | Track commanded $v_x$ velocity. | 1.0 |
| tracking_lin_vel_y | Track commanded $v_y$ velocity. | 1.0 |
| tracking_ang_vel | Track commanded yaw rate. | 2.0 |
| base_height | Reward proximity to target height (0.65m). | 0.2 |
| orientation | Penalize torso tilt from vertical. | -5.0 |
| feet_swing | Reward proper foot swing phase. | 3.0 |
| feet_slip | Penalize slipping contacts. | -0.1 |
| feet_yaw_diff | Penalize difference in foot yaw angles. | -1.0 |
| feet_yaw_mean | Penalize deviation from base yaw. | -1.0 |
| feet_roll | Penalize foot roll angles. | -1.0 |
| feet_distance | Penalize feet being too close. | -10.0 |

**Reward terms (simplified for transfer):**

**Disabled reward terms:** torques, torque_tiredness, power, lin_vel_z, ang_vel_xy, dof_vel, dof_acc, root_acc, action_rate, dof_pos_limits, collision, feet_vel_z (set to 0.0 scale).

**Implementation details**

- External pushes enabled (interval: 5.0-10.0s, magnitude: 0.1-1.0N)
- Normalized observations ensure consistent scaling across simulators

Table 7: Domain randomization parameters (Note: $U[a, b]$ denotes uniform distribution over $[a, b]$, $\pm U(c)$ denotes uniform distribution over $[-c, c]$)

| Parameter | Range | Description |
|-----------|-------|-------------|
| Floor friction | $U[0.2, 0.6]$ | Ground friction coefficient |
| Joint friction loss | $\times U[0.9, 1.1]$ | Joint friction scaling |
| Joint armature | $\times U[1.0, 1.05]$ | Joint armature scaling |
| Link masses | $\times U[0.98, 1.02]$ | Body mass scaling |
| Torso mass | $+U[-1.0, 1.0]$ kg | Additional torso mass |
| Initial joint positions | $\pm U[0.05]$ rad | Default pose randomization |
| Joint stiffness | $\times U[0.7, 1.3]$ | Actuator gain scaling |
| Joint damping | $\times U[0.7, 1.3]$ | Damping coefficient scaling |
| Ankle damping | $\times U[0.5, 2.0]$ | Higher range for ankle joints |

# D  LARGE LANGUAGE MODEL USAGE

We gratefully acknowledge the use of ChatGPT and DeepSeek to assist in polishing, refining, and condensing the text of this paper.

