# OpenReview forum: "Towards Bridging the Gap between Large-Scale Pretraining and Efficient Finetuning for Humanoid Control"
_ICLR.cc/2026/Conference — ICLR 2026 Poster_

### Official Review · Reviewer_2rhQ · 2025-10-27

**Soundness:** 3
**Presentation:** 2
**Contribution:** 3
**Rating:** 6
**Confidence:** 3

**Summary:**

The authors address issues with reinforcement learning control of humanoid robotic systems. They point out that proximal policy optimization can be brittle when transferred to zero-shot settings and is sample-inefficient even in parallel environments. The proposed solution is to use the soft actor-critic algorithm with a learned physics-aware world model to first pre-train a policy on a large-scale, parallelizable simulator and then finetune online using synthetic samples generated by the world model for the target humanoid task. The authors demonstrate that their pretrained policy can be deployed zero-shot in the real world and performs better during finetuning on simulation tasks.

**Strengths:**

> The authors provide a fast, scalable implementation of SAC, which opens up new avenues for research on the limitations of online reinforcement learning
> The authors conduct extensive studies to verify the efficacy of their algorithm. I appreciated their use of 8 seeds, which is much better than the typical practice in other deep RL research (as few as three seeds in most works I have reviewed). The inclusion of zero-shot transfer in real robotics was also a promising finding, although no experiments were conducted to finetune on real-robot data.
> There are interesting findings to this work, such as identifying the limitations of prior research on plasticity, finding that the pretraining algorithm does affect finetuning performance.

**Weaknesses:**

Our biggest concern is the quality of organization and writing in the current submission. A good example of this is the exclusion of results to answer Q3 in the experiment section. If the paper had reduced content in other sections, it would have been easier to include them in the main paper. For example, Comments on design choices in the method section can be moved to the appendix, such as those about the experimental evidence used to justify decisions, which should be explained in the Q3 results (lines 256 & 257). The introduction could be more coherent and more transparent about why physics-aware world models are being employed.

In terms of contribution, this type of work might have more impact at a robotics-focused conference. The authors' method seems promising as a pretrain-and-finetune solution for humanoids, but it offers limited novelty beyond addressing issues with SSRL, which their work is heavily influenced by. There are aspects of the work that challenge this comment, such as their finding that issues of plasticity (lines 223-227) do not arise when scaling the parallel environments, but this is not emphasized in the main experiments and is left as future work.

The experiment section demonstrates that the framework works and identifies which hyperparameters the model is sensitive to. I did not find the results shown for Q1 particularly insightful, because all that is shown is that the authors do "just as well" as other methods. If there are other benefits, this should be emphasized; otherwise, Q1 could be moved to the appendix or condensed to make space for more critical analysis of LIFT. Q2 & Q3 deserve more discussion or consideration, particularly as Q2 is the application scenario the authors are studying, and Q3 helps the reader understand which hyperparameters are essential for applying LIFT.



Writing opinions

Line 063: The last sentence in the opening paragraph doesn't logically follow from the limitations of PPO per se, particularly because the "pretrain-finetune paradigm" isn't explained as to WHY it solves PPO issues. Maybe something along the lines of "off-policy algorithms are a means of addressing this" would make more sense given the following paragraph.
Line 065: The author uses "However" twice in this paragraph.

Line 076: "we conduct a preliminary study in which humanoid walking is trained from scratch while limiting data collection to deterministic execution."
> This sentence should be deleted. When I read this in the introduction, my first question was whether the research was complete. That's probably not what the authors meant, but that is how it came across.

Line 082: (ii) The use of physics-informed world models is not motivated well in the introduction, so from the reader's perspective, this comes out of nowhere as to why these are used over non-physics-aware models.

Line 238: The information on the state vector could be moved to the appendix.

**Questions:**

Q0: Could the author's expand on the following interpretation (Line 416): . PPO maintains reasonable performance initially but degrades over time, eventually collapsing, likely due to its KL-regularized updates that limit fast policy adaptation? Did the author's include the KL divergence between the current policy and old policy? What about the clip coefficient?

Q1: For Q1, if your algorithm only does just as well as baselines, what inherent advantage does LIFT provide that prior methods do not have for the pretraining phase? I'm thinking about algorithmic benefits, not the transfer learning aspect (i.e. Q2).

Q2: Line 211 - Are the authors actually using a Gaussian policy, or is there a tanh activation transformation to normalize actions? This transformed distribution was used in the original SAC.

Q3: How much of the world model is learned as opposed to being defined by the BRAX? What simulator is used to pretrain the model? Is it re-using the same tools as the world model simulator? Is there any issue of mismatch between the pre-trained simulator and the target environment?

Q4: Why is SSRL not included as a comparison in the reported results of the paper for Q1 or Q2? The author's algorithm appears heavily motivated by this prior work, yet no results appear in the main paper.

Q5: What are the reasons for not comparing LIFT with a physics-informed model vs a non-physics-informed model?

Q6: What are the potential limitations of LIFT for high-dimensional observation data? What changes would be needed to use images, for example?

---

> ### Author Response · Authors · 2025-11-26
> **Official Reply to Reviewer JXM1 (1/5)**
>
> Thank you very much for your thorough review and valuable suggestions. In response to each of your points, we have provided detailed replies and made corresponding adjustments to our paper.
>
> > (Weakness #1) Our biggest concern is the quality of organization and writing in the current submission. A good example of this is the exclusion of results to answer Q3 in the experiment section. If the paper had reduced content in other sections, it would have been easier to include them in the main paper. For example, Comments on design choices in the method section can be moved to the appendix, such as those about the experimental evidence used to justify decisions, which should be explained in the Q3 results (lines 256 & 257). The introduction could be more coherent and more transparent about why physics-aware world models are being employed.
>
> Thank you very much for the valuable suggestions. In response, we have made the following revisions:
>
> 1. We move most of the Q3 experiments from the appendix into the main paper Section 5.3 (Ablation Study), including the analyses on effect of pretraining and comparison between physics-informed vs. non-physics-informed world models.
>
> 2. We incorporated the SSRL baseline into the Q2 experiments and explicitly included the results in the Fig. 2. However, the original SSRL implementation does not directly support humanoid training in Brax: the policy fails to enter a stable learning regime and repeatedly resets due to mismatches between the proprioceptive state used in SSRL and Brax’s generalized state representation. For example, the global angular velocity in the proprioceptive state is incorrectly interpreted as the local angular velocity expected by Brax, and the privileged state omits base height, which is crucial for accurate world-model rollouts during finetuning. For quadrupeds, assuming a roughly fixed base height can be a tolerable approximation because body height varies only mildly. For humanoids, base height can vary substantially, which breaks down this approximation. After identifying and correcting these issues, SSRL begins to show an initial trend toward convergence. We view these adjustments as practical engineering steps needed to ensure a fair comparison (Evaluating only the unmodified SSRL code would mainly reflect implementation mismatches rather than algorithmic differences), and they also reflect part of our contribution in extending existing frameworks to humanoid platforms. Even with these corrections, SSRL still fails to converge on high-speed tracking tasks (1.0–1.5 m/s), as shown in Section 5.2 (Finetuning Experiments), highlighting the difficulty of training humanoid locomotion from scratch and further motivating our pretrain–finetune design. For this reason, we describe these engineering details briefly in the Methods Section rather than in the main experimental analysis.
>
> 3. We revised the introduction to more clearly explain the motivation for using physics-informed world models.

---

> ### Author Response · Authors · 2025-11-26
> **Official Reply to Reviewer JXM1 (2/5)**
>
> > (Weakness #2)  In terms of contribution, this type of work might have more impact at a robotics-focused conference. The authors' method seems promising as a pretrain-and-finetune solution for humanoids, but it offers limited novelty beyond addressing issues with SSRL, which their work is heavily influenced by. There are aspects of the work that challenge this comment, such as their finding that issues of plasticity (lines 223-227) do not arise when scaling the parallel environments, but this is not emphasized in the main experiments and is left as future work.
>
> In our updated Q2 experiments, we show that SSRL does not scale effectively to more complex, higher-DoF humanoid systems: in Brax, the original SSRL pipeline either fails to learn meaningful locomotion or progresses extremely slowly, even after careful engineering adaptations. This negative result is, in itself, informative—it highlights that straightforwardly applying existing model-based methods for sample-efficient humanoid learning is not sufficient. Our work takes a different direction and proposes a complete pretrain–finetune framework specifically designed to make humanoid adaptation both fast and sample-efficient. Our revised manuscript additionally includes real-world finetuning experiments on a physical humanoid, demonstrating that the full pipeline can be deployed on hardware. (Some **video demos** can be seen in: [real-world-finetuning](https://lift-humanoid.github.io/#:~:text=Real%2Dworld%20Finetuning%20in%20Booster%20T1)) Overall, our goal is not merely to “fix” SSRL, but to develop and validate a framework that allows humanoid robots to adapt efficiently with limited data from new simulators or real robots.
>
> Regarding the reviewer’s comment on plasticity: we did not emphasize this aspect in the main experiments because, in our current implementation, large-scale parallel environment training already mitigates the plasticity issues reported in prior work. This setup is sufficient for pretraining humanoid policies and achieving reliable zero-shot transfer to the real robot. Our primary goal, however, is to enable continued finetuning on the humanoid platform. We expect that revisiting these plasticity-related design tricks may become important when scaling to more complex tasks, such as the [whole-body motion tracking experiments](https://lift-humanoid.github.io/#:~:text=with%20whole%20body-,tracking,-pipeline%20on%20the) we added during the rebuttal phase, where faster and more stable convergence is required.
>
>
> > (Weakness #3)  The experiment section demonstrates that the framework works and identifies which hyperparameters the model is sensitive to. I did not find the results shown for Q1 particularly insightful, because all that is shown is that the authors do "just as well" as other methods. If there are other benefits, this should be emphasized; otherwise, Q1 could be moved to the appendix or condensed to make space for more critical analysis of LIFT. Q2 & Q3 deserve more discussion or consideration, particularly as Q2 is the application scenario the authors are studying, and Q3 helps the reader understand which hyperparameters are essential for applying LIFT.
>
> 1. We have moved the full experimental curves for Q1 to the Appendix A.2. The purpose of Q1 is to demonstrate that our framework achieves comparable wall-clock pretraining performance to strong baselines such as PPO. This result is important: if pretraining required days or weeks—as is common in traditional model-based RL—then the framework would be impractical for the robotic community regardless of its fine-tuning ability.
>
> 2. We have moved most of the Q3 experimental results from the appendix into the main paper and for Q2, we expanded Section 5.2 (Finetuning Experiments) to include the results and discussions of the real-world fine-tuning experiments.

---

> ### Author Response · Authors · 2025-11-26
> **Official Reply to Reviewer JXM1 (3/5)**
>
> > (Writing opinions #1) Line 063: The last sentence in the opening paragraph doesn't logically follow from the limitations of PPO per se, particularly because the "pretrain-finetune paradigm" isn't explained as to WHY it solves PPO issues. Maybe something along the lines of "off-policy algorithms are a means of addressing this" would make more sense given the following paragraph. Line 065: The author uses "However" twice in this paragraph.
>
> Thank you for the suggestion. We have revised the introduction.
>
> > (Writing opinions #2) Line 076: "we conduct a preliminary study in which humanoid walking is trained from scratch while limiting data collection to deterministic execution." This sentence should be deleted. When I read this in the introduction, my first question was whether the research was complete. That's probably not what the authors meant, but that is how it came across.
>
> Thank you for pointing this out. Our intention was to refer to the preliminary experiments reported in Appendix A.1, not to imply that the main research was incomplete. To avoid this unintended interpretation, we have removed the sentence from the introduction.
>
> > (Writing opinions #3) Line 082: (ii) The use of physics-informed world models is not motivated well in the introduction, so from the reader's perspective, this comes out of nowhere as to why these are used over non-physics-aware models.
>
> Thank you for the suggestion. We have revised the introduction to provide a clearer motivation for using physics-informed world models.
>
> > (Writing opinions #4) Line 238: The information on the state vector could be moved to the appendix
>
> We have condensed this section while retaining the essential description in the main paper, as it directly addresses concerns raised by reviewer JXM1 regarding whether external privileged information (e.g., ground friction) is used during world-model training. We clarify that our method relies solely on the robot’s proprioceptive state and does not use any additional non-robot privileged information during either pretraining or finetuning.
>
> > (Question #0) Could the author's expand on the following interpretation (Line 416): . PPO maintains reasonable performance initially but degrades over time, eventually collapsing, likely due to its KL-regularized updates that limit fast policy adaptation? Did the author's include the KL divergence between the current policy and the old policy? What about the clip coefficient?
>
> We apologize for the confusion caused by our earlier wording. The PPO baseline used in our experiments is the official Brax implementation, which relies on the clipped surrogate objective and does not include any explicit KL-regularization term or adaptive KL penalties. We use a clipping coefficient of 0.2. We have updated the description in the paper as follows:
>
> “Although PPO’s clipping mechanism stabilizes updates by keeping the new policy close to the previous one, its performance in our setting initially remains reasonable but then gradually degrades and ultimately collapses. This suggests that, under deterministic execution and limited data collection, PPO struggles to sustain stable policy improvement.”
>
>
> > (Question #1) For Q1, if your algorithm only does just as well as baselines, what inherent advantage does LIFT provide that prior methods do not have for the pretraining phase? I'm thinking about algorithmic benefits, not the transfer learning aspect (i.e. Q2).
>
> If only consider the pretraining stage in isolation, our current large-scale SAC did not have advantages over existing algorithms including PPO and FASTD3. Our intention with Q1 is to show that LIFT’s SAC pretraining can match strong baselines such as PPO and FastTD3 in wall-clock efficiency and final performance. Meanwhile it supports us to "do more" which is purpose-built to be extended by a physics-informed world model and a finetuning loop. In other words, Q1 establishes that LIFT pretraining is competitive with strong baselines as a standalone solution, while Q2 shows the additional benefit of our design: the same pretraining stage serves as a foundation for sample-efficient adaptation (finetuning) that prior methods (including PPO and SSRL) do not provide for humanoids.
>
> > (Question #2) Line 211 - Are the authors actually using a Gaussian policy, or is there a tanh activation transformation to normalize actions? This transformed distribution was used in the original SAC.
>
>  We use the standard tanh-transformed Gaussian policy, following the original SAC formulation.

---

> ### Author Response · Authors · 2025-11-26
> **Official Reply to Reviewer JXM1 (4/5)**
>
> >(Question #3) How much of the world model is learned as opposed to being defined by the BRAX? What simulator is used to pretrain the model? Is it re-using the same tools as the world model simulator? Is there any issue of mismatch between the pre-trained simulator and the target environment?
>
> In our framework, most of the dynamics are provided by Brax, while the right two residual components on the right-hand side of Eq. (2) are learned. Concretely, we use Brax’s differentiable rigid-body primitives for the mass matrix, Coriolis/centrifugal terms, and gravity, and we apply a PD controller to map actions to motor torques. On top of it, a neural network as shown in Eqs. (3)(4) predicts only the residual external/dissipative torques and the prediction uncertainty of the next state. Thus, the world model is largely defined by the known rigid-body dynamics in Brax, with learning focused on contact and other unmodeled effects.
>
> Policy pretraining is performed in MuJoCo Playground, using thousands of parallel environments, and all transitions from this stage are logged. The world model itself (the residual neural network) is then pretrained offline using these logged transitions, but its internal dynamics are implemented entirely in Brax.
>
> Besides that we align the state representation, coordinate frames, quaternion conventions, observation normalization, and PD controller settings between MuJoCo Playground and Brax, the mujoco playground does not share any same tool with our world model simulator or brax simulator. However, the two simulators differ in contact modeling and constraints, so successful transfer requires LIFT to adapt across nontrivial dynamics differences. This sim-to-sim gap is in fact an intentional part of our evaluation. Brax uses a differentiable, penalty-based (visco-elastic spring–damper) contact model to keep the dynamics differentiable. In contrast, MuJoCo uses an implicit time-stepping solver with constraint-based contacts formulated as an optimization problem, leading to short, sharp contact events and tight enforcement of joint and equality constraints.
>
> >(Question #4) Why is SSRL not included as a comparison in the reported results of the paper for Q1 or Q2? The author's algorithm appears heavily motivated by this prior work, yet no results appear in the main paper.
>
> We included SSRL in the original appendix to analyze the effect of pretraining, and during the rebuttal period we conducted additional experiments to systematically compare it in Q2. The updated Q2 results are now incorporated into the main paper Section 5.2 (Finetuning Experiments).
>
> We do not include SSRL in Q1 because the original SSRL framework does not support large-scale pretraining. Extending SSRL to handle thousands of parallel environments for world-model training leads to severe GPU memory issues. Even when the number of environments is reduced to only a few dozen, the training becomes too slow to yield meaningful Q1 performance on 1024 parallel evaluation environments within a practical one-hour wall-clock budget. Effectively enabling pretraining in this setting would require developing a new model-based RL variant of SSRL beyond its original design.
>
> >(Question #5)  What are the reasons for not comparing LIFT with a physics-informed model vs a non-physics-informed model?
>
> In fact, our comparison between LIFT and MBPO in the original submission is precisely the comparison between a physics-informed and a non-physics-informed world model. This training curve was originally placed in the appendix; in the revised version, we have moved it to Section 5.3 (Ablation Study) and highlighted it.
>
> We pretrain MBPO's ensemble world model on the same dataset and finetune with identical hyperparameters to LIFT; the only difference is the choice of world model. MBPO fails to converge: episode return remain near zero. On the test set its mean squared error (MSE) of world model is substantially worse than LIFT's. During model-based rollouts, a stochastic policy frequently produces actions that lie outside the distribution that has been seen in world-model training. These out-of-distribution actions induce physically implausible predictions (e.g., body height) in MBPO rollouts, which cause the critic loss to explode and inhibit policy improvement. This behavior likely stems from the purely neural network's limited ability to generalize.
>
> By contrast, LIFT's physics-informed world model supplies strong inductive priors that improve generalization under limited data and produce stable, learnable rollouts, enabling successful finetuning.

---

> ### Author Response · Authors · 2025-11-26
> **Official Reply to Reviewer JXM1 (5/5)**
>
> > (Question #6) What are the potential limitations of LIFT for high-dimensional observation data? What changes would be needed to use images, for example?
>
> At present, LIFT operates purely on proprioceptive state and does not take camera or tactile sensor as input. In future work, as we scale LIFT to tasks with vision-based objectives—such as dexterous manipulation—latent world models like DreamerV3 [1] may become necessary to capture dynamics beyond the robot’s own state. A potential direction is to integrate LIFT’s rigid-body world model with a latent dynamics model: external high-dimensional signals can be encoded into a latent space using a multi-modal encoder (as in DreamerV3), while predictions from LIFT’s rigid-body dynamics (e.g., base position and velocity) serve as structured inputs to the latent dynamics. The decoder then reconstructs the next high-dimensional sensory observation. Conversely, the learned latent state may also provide auxiliary information that helps improve the prediction of the robot’s rigid-body dynamics. This hybrid approach, akin to the structure-aware latent modeling explored in [2], may allow latent dynamics to complement physically grounded dynamics, enabling LIFT to scale to tasks requiring external perception and richer environmental interaction. We have added a discussion of high-dimensional external sensor in the revised manuscript Section 6 (Conclusion and Discussion).
>
> [1] Hafner D, Pasukonis J, Ba J, et al. Mastering diverse domains through world models[J]. arXiv preprint arXiv:2301.04104, 2023.
>
> [2] Greydanus S, Dzamba M, Yosinski J. Hamiltonian neural networks[J]. Advances in neural information processing systems, 2019, 32.

---

### Official Review · Reviewer_JXM1 · 2025-10-31

**Soundness:** 2
**Presentation:** 3
**Contribution:** 3
**Rating:** 6
**Confidence:** 3

**Summary:**

This paper introduces LIFT, a three-stage framework that bridges large-scale off-policy pretraining and efficient safe finetuning for humanoid robot locomotion. The key pipeline involves: (i) large-scale pretraining of policies using Soft Actor-Critic (SAC) in massively parallel simulators, (ii) pretraining a physics-informed world model leveraging Lagrangian dynamics and residual neural predictions on stored transitions, and (iii) iteratively finetuning both the policy and world model in new environments using deterministic real-world execution and stochastic exploration confined to the model. Experiments on multiple humanoids and transfer scenarios indicate that LIFT achieves competitive pretraining performance and robust, efficient finetuning—outperforming prior model-free and model-based RL baselines under challenging adaptation conditions. The code is open-sourced.

**Strengths:**

- The proposed pipeline is systematically motivated, decomposing the challenge of bridging data-hungry RL pretraining with safety and efficiency in real-world adaptation.
- The integration of Lagrangian dynamics with learned residuals for improved world model rollouts is novel in the humanoid domain and addresses both sample efficiency and stability concerns.
- The work provides results for two complex humanoid platforms (Booster T1 and Unitree G1), encompassing flat and rough terrain, various velocity targets (in-distribution, long-tail, and out-of-distribution), and sim-to-sim as well as sim-to-real transfers. Notably, the policy achieves zero-shot deployment and robust adaptation.

**Weaknesses:**

- **Task Diversity**: The evaluation focuses on forward locomotion tasks (varying speed targets on flat or rough terrain). This is a narrow slice of humanoid skills. The significance would be further bolstered by testing, say, different locomotion gaits or disturbances, or tasks like turning, obstacle avoidance, etc. It’s unclear how readily the approach extends to non-locomotion tasks or more complex objectives.
- **Use of Privileged Information**: The method relies on a privileged state (including full physical state and other quantities like ground friction) for training the world model and critic. This is standard in sim-to-real pipelines, but it presumes access to those privileged observations. The paper doesn’t fully discuss how feasible this is outside simulation – e.g., obtaining exact friction or global state on a real robot might require estimation. If such info is unavailable or inaccurate, it could affect performance. The authors do not address this point in depth, and it could be a concern for deploying the finetuning part in the wild.
- **Comparison to Other Model-Based RL**: The paper compares to MBPO as a baseline for learned dynamics, but other model-based approaches (e.g., Dreamer or recent model-based fine-tuning methods) are not explicitly compared in finetuning. Given that DreamerV3 and others have shown strong model-based learning on locomotion, an empirical comparison or at least a discussion would have been useful to position LIFT against the broader landscape of model-based RL in terms of performance and efficiency.

**Questions:**

1. Can the authors clarify their design choices regarding the length and scaling of the autoregressive unroll horizon in world model training? How sensitive is LIFT to this hyperparameter in practice?
2. In real-robot adaptation, what additional safety or recovery mechanisms (beyond physics-consistent reset) would be paramount, and how does LIFT plan to address distributional shift that may be more severe than sim-to-sim transfer?
3. Is there a plan or preliminary evidence for multi-task or non-locomotion humanoid adaptation under LIFT, or does the framework generalize only to velocity-tracking locomotion?

---

> ### Author Response · Authors · 2025-11-26
> **Official Reply to Reviewer JXM1 (1/3)**
>
> Thank you very much for your thorough review and valuable suggestions. In response to each of your points, we have provided detailed replies and made corresponding adjustments to our paper.
>
> > (Weakness #1) Task Diversity: The evaluation focuses on forward locomotion tasks (varying speed targets on flat or rough terrain). This is a narrow slice of humanoid skills. The significance would be further bolstered by testing, say, different locomotion gaits or disturbances, or tasks like turning, obstacle avoidance, etc. It’s unclear how readily the approach extends to non-locomotion tasks or more complex objectives.
>
> Our framework is built around learning a world model of the robot’s state dynamics, so any task whose reward depends only on the robot proprioceptive state can, in principle, be handled directly. This includes different locomotion gaits, whole-body motion tracking, and balance under disturbances, since as long as the reward can be computed from the state, the policy can be finetuned entirely through world-model rollouts.
>
> To better demonstraste this potential, during the rebuttal period we extended our pretraining pipeline from velocity-tracking locomotion to BeyondMimic-style whole-body tracking [1]. Specifically, we reimplemented the observation and reward structure of BeyondMimic in JAX within MuJoCo Playground and used the Unitree motion dataset (LAFAN1) to pretrain a tracking policy for the Unitree G1 humanoid. Updated video demonstrations are provided on our [project website](https://lift-humanoid.github.io/#:~:text=with%20whole%20body-,tracking,-pipeline%20on%20the). Due to engineering effort (reproducing identical observation/reward setups and contact handling across MuJoCo and Brax, and integrating this into our world-model finetuning loop), we were only able to complete the pretraining stage within the rebuttal timeframe. We therefore treat these whole-body tracking results as preliminary evidence of LIFT’s broader potential. And we explicitly  discuss this in Appendix A.7 (Non-locomotion tasks) of the revised manuscript. This limitation is practical, not conceptual: the same stage-(iii) finetuning procedure applies once the corresponding Brax environment is ready or we have access to the real hardware of the Unitree G1 robot. Some video demos can be seen in: [https://lift-humanoid.github.io/#:~:text=with%20whole%20body-,tracking,-pipeline%20on%20the](https://lift-humanoid.github.io/#:~:text=with%20whole%20body-,tracking,-pipeline%20on%20the)
>
> Regarding disturbances, our world model predicts both residual contact/dissipative terms and the predictive uncertainty of the next state. When external pushes occur, the model cannot fully explain these transitions from proprioceptive signals alone and therefore assigns high uncertainty to them. During finetuning, this uncertainty is injected into model rollouts, effectively exposing the policy to disturbance-like transitions and encouraging robust recovery. In pretraining, we also injected random pushes (0.1–1.0 m/s) to Unitree G1 and Booster T1, and we observe that these data improve downstream finetuning performance, as shown in the figure 4 in the Section 5.3 (Ablation Study-Effect of Pretraining).
>
> For tasks involving external objects (e.g., kicking a ball), the current framework would need to be extended to also model object dynamics, for example by augmenting the state with object pose and using simple physical priors such as conservation of momentum. This is feasible but requires additional engineering and environment design, so we explicitly leave it as future work. Likewise, obstacle avoidance and more complex navigation are more naturally handled at a higher level: once a reliable low-level tracking controller is available (as in our setup), a high-level policy can be trained to output velocity or pose targets, potentially with its own high-level world model as explored in hierarchical model-based RL [2]. We have added a short discussion along these lines in Appendix A.7 (Non-locomotion tasks) and now make it clearer that extending LIFT to hierarchical and object-centric tasks is a promising direction, but beyond the scope of this paper.
>
> References:
>
> [1] Liao Q, Truong T E, Huang X, et al. Beyondmimic: From motion tracking to versatile humanoid control via guided diffusion[J]. arXiv preprint arXiv:2508.08241, 2025.
>
> [2] Hansen N, SV J, Sobal V, et al. Hierarchical world models as visual whole-body humanoid controllers[J]. arXiv preprint arXiv:2405.18418, 2024.

---

> ### Author Response · Authors · 2025-11-26
> **Official Reply to Reviewer JXM1 (2/3)**
>
> > (Weakness #2) Use of Privileged Information: The method relies on a privileged state (including full physical state and other quantities like ground friction) for training the world model and critic. This is standard in sim-to-real pipelines, but it presumes access to those privileged observations. The paper doesn’t fully discuss how feasible this is outside simulation – e.g., obtaining exact friction or global state on a real robot might require estimation. If such info is unavailable or inaccurate, it could affect performance. The authors do not address this point in depth, and it could be a concern for deploying the finetuning part in the wild.
>
> Thank you for raising this concern. We clarify that our use of the term privileged information referred only to the standard asymmetric actor–critic formulation, where the critic may receive additional state features compared to the actor. In the original text, this wording unintentionally suggested that we used environmental parameters such as ground friction; however, our experiments do not use any non-robot or environment-specific privileged information. We have corrected this in the revised paper.
>
> Our method requires only the robot’s own state: base height, base orientation, joint positions, joint velocities, and base linear and angular velocities. Factors such as ground friction are not provided explicitly and are implicitly captured by the contact and dissipative terms predicted by our world model (Eq. (3)).
>
> During the rebuttal phase, we also conducted real-world finetuning experiments. These experiments do not rely on access to ground friction or other environmental parameters—only the robot-state information listed above. Currently, the Booster T1 platform does not provide base-height estimation, so we used a Vicon system for this purpose, which restricts experiments to indoor environments. This limitation is discussed in the revised manuscript Section 5.2 (Finetuning Experiments) and Section 6 (Conclusion and Discussion), and we plan to explore onboard sensing (e.g., a small onboard camera for base-height estimation) to enable finetuning in the wild. A video demonstration of real-world finetuning is available: [project website](https://lift-humanoid.github.io/#:~:text=Real%2Dworld%20Finetuning%20in%20Booster%20T1)
>
> > (Weakness #3)Comparison to Other Model-Based RL: The paper compares to MBPO as a baseline for learned dynamics, but other model-based approaches (e.g., Dreamer or recent model-based fine-tuning methods) are not explicitly compared in finetuning. Given that DreamerV3 and others have shown strong model-based learning on locomotion, an empirical comparison or at least a discussion would have been useful to position LIFT against the broader landscape of model-based RL in terms of performance and efficiency.
>
> Thank you for the suggestion. We have added a discussion of DreamerV3 and latent-space world models in the revised manuscript Section 6 (Conclusion and Discussion). At present, LIFT operates purely on proprioceptive state and does not take camera or other vision-based observations as input, unlike vision-centric frameworks such as DreamerV3. In future work, as we scale LIFT to tasks with vision-based objectives—such as dexterous manipulation—latent world models like DreamerV3 may become necessary to capture dynamics beyond the robot’s own state. A potential direction is to integrate LIFT’s rigid-body world model with a latent dynamics model: external high-dimensional signals can be encoded into a latent space using a multi-modal encoder (as in DreamerV3), while predictions from LIFT’s rigid-body dynamics (e.g., base position and velocity) serve as structured inputs to the latent dynamics. The decoder then reconstructs the next high-dimensional sensory observation. Conversely, the learned latent state may also provide auxiliary information that helps improve the prediction of the robot’s rigid-body dynamics. This hybrid approach, akin to the structure-aware latent modeling explored in [3], may allow latent dynamics to complement physically grounded dynamics, enabling LIFT to scale to tasks requiring external perception and richer environmental interaction.
>
> We did not include DreamerV3 in our finetuning comparisons for two reasons. First, existing Dreamer-style methods do not support the large-scale parallelized robot training required in our pretraining stage, making a fair and controlled comparison difficult. Second, non-physics-informed model-based RL algorithms perform poorly under our deterministic data-collection setting and are unable to converge within 40k environment steps, which is the finetuning budget used in our experiments. We compare with MBPO because it can share the same SAC-based actor–critic pretraining and a similar world-model pretraining procedure as ours.
>
> References:
>
> [3] Greydanus S, Dzamba M, Yosinski J. Hamiltonian neural networks[J]. Advances in neural information processing systems, 2019, 32.

---

> ### Author Response · Authors · 2025-11-26
> **Official Reply to Reviewer JXM1 (3/3)**
>
> > (Question #1) Can the authors clarify their design choices regarding the length and scaling of the autoregressive unroll horizon in world model training? How sensitive is LIFT to this hyperparameter in practice?
>
> Autoregressive world-model training has been widely used in prior model-based RL methods [4–5], and has been shown to improve predictive accuracy. Intuitively, forcing the model to predict multiple steps encourages it to produce more accurate one-step predictions, which in turn leads to higher-quality synthetic data for policy learning.
>
> In practice, LIFT is not highly sensitive to the choice of autoregressive unroll horizon within a reasonable range. In our experiments, we find that increasing the horizon to 2 is sufficient to yield consistent benefits. Further increasing the horizon significantly increases computational cost but does not provide meaningful performance gains.
>
> > (Question #2)  In real-robot adaptation, what additional safety or recovery mechanisms (beyond physics-consistent reset) would be paramount, and how does LIFT plan to address distributional shift that may be more severe than sim-to-sim transfer?
>
>  In real-robot adaptation, our experiments employ strict safety and termination mechanisms. An episode is immediately stopped whenever the base height, global base linear velocity, base Euler angles, joint positions, or joint velocities exceed predefined safety thresholds. Human operators also remotely terminate any episode in which the robot exhibits unsafe behavior, such as drifting toward obstacles.
>
> Once a termination condition is triggered, we immediately switch the Booster T1 into damping mode, which constrains joint motion and reduces risk. After the robot stabilizes, we use the platform’s built-in walking controller to guide it back to the starting area, and then apply the default standing controller to return it to a safe preparation posture before the next data-collection cycle.
>
>
> We've included all those real-world fine-tuning results and discussions in the revised manuscript Section 5.2 (Finetuning Experiments) and Section 6 (Conclusion and Discussion).
>
> > (Question #3) Is there a plan or preliminary evidence for multi-task or non-locomotion humanoid adaptation under LIFT, or does the framework generalize only to velocity-tracking locomotion?
>
> Yes. We do plan to extend LIFT to multi-task and non-locomotion humanoid adaptation, and we have added this discussion to the Appendix A.7 (NON-LOCOMOTION TASKS). During the rebuttal period, we conducted preliminary experiments showing that the pretraining stage of LIFT can already support whole-body motion tracking, as demonstrated on our [project website](https://lift-humanoid.github.io/#:~:text=with%20whole%20body-,tracking,-pipeline%20on%20the). Thus, the framework is not limited to velocity-tracking locomotion; in principle, it can be applied directly to more complex tasks such as full-body tracking, as discussed in Response to Weakness 1.
>
> Regarding the multi-task setting, once we pretrain LIFT policy on a large-scale motion dataset for whole-body tracking, we will obtain a diverse, multi-task dataset for world-model pretraining (similar in spirit to TDMPC2[6]). This opens the possibility of training a multi-task world model that can then be finetuned on different downstream tasks directly on the physical humanoid robot.
>
> References:
>
> [4] Li C, Krause A, Hutter M. Robotic world model: A neural network simulator for robust policy optimization in robotics[J]. arXiv preprint arXiv:2501.10100, 2025.
>
> [5] Levy J, Westenbroek T, Fridovich-Keil D. Learning to walk from three minutes of real-world data with semi-structured dynamics models[J]. arXiv preprint arXiv:2410.09163, 2024.
>
> [6] Hansen N, Su H, Wang X. Td-mpc2: Scalable, robust world models for continuous control[J]. arXiv preprint arXiv:2310.16828, 2023.

---

### Official Review · Reviewer_2CYj · 2025-11-01

**Soundness:** 3
**Presentation:** 3
**Contribution:** 3
**Rating:** 6
**Confidence:** 3

**Summary:**

This paper proposes LIFT, a three-stage framework to bridge the gap between large scale pre-training and sample-efficient fine-tuning for humanoid locomotion control.

The framework first challenges the consensus that PPO is the default for large-scale training. It demonstrates that off-policy SAC, when implemented in JAX with large-batch updates and a high Update-To-Data (UTD) ratio, can achieve comparable or faster wall-clock convergence than PPO for pre-training, while also enabling zero-shot sim-to-real.

The authors then propose a model-based fine-tuning stage, where a physics-informed world model (combining Lagrangian dynamics with a learned residual network) is pre-trained on the SAC simulation data. In stage (iii), this world model is adapted to a new environment. The robot collects rollout data in the target environment using a deterministic policy for safety. This new data is used to fine-tune the world model. Stochastic exploration is then confined to rollouts within this updated world model, and the policy is updated using this safe, synthetic data.

The authors validate this approach in "sim-to-sim" experiments, showing that the LIFT framework can successfully fine-tune policies for new, out-of-distribution tasks (e.g., new velocity commands in a new simulator) where model-free fine-tuning of SAC, PPO, and FastTD3 all fail.

**Strengths:**

* The paper provides a valuable finding by demonstrating that SAC, an off-policy algorithm, can be successfully scaled for massively parallel pre-training of humanoid controllers.  The open-sourcing of a JAX-based SAC implementation for this purpose is a welcome contribution.
* The proposed hybrid physics-informed world model works and the target environment adaptation works well for humanoid simulation. The ablation study in Appendix A.3 (Fig. 6) shows that a standard "black-box" MBPO-style model completely fails to adapt, while LIFT's physics-informed model succeeds. This provides strong evidence that embedding physics priors is essential for world-model-based adaptation in this domain.
* The LIFT framework's main contribution is its integration of pre-training with a model-based fine-tuning loop. The proposed world-model finetuning framework could be used in more humanoid finetuning cases.

**Weaknesses:**

* Lack of Real-World Fine-Tuning (Sim-to-Sim): The paper's primary weakness is that its core claim—safe, efficient fine-tuning—is only validated in a "sim-to-sim" setting (MuJoCo to Brax). While the paper shows zero-shot *pre-training* on a real robot (Appendix A.6), it does not "close the loop" and test the *fine-tuning* procedure in the real world. The claim of "safety" is significantly weaker without this, as collecting even "deterministic" data in a new real-world environment carries risks (e.g., from initial state distribution mismatch or model error) that the paper does not address.

* Missing Discussion on Safety for Real-World Adaptation: Following W1, the paper overlooks the practical hardware challenges of data collection for fine-tuning. Even a deterministic policy can fail. Methods like RTR [1] have shown that specially designed safety systems (e.g., a "teacher" robot arm) are crucial for enabling this kind of real-world adaptation on humanoids. A discussion of these practical requirements is missing.

* Incomplete Related Work and Missing Citations: The related work section is sparse in several key areas, weakening the paper's positioning and claims of novelty. (a) Context for SAC Pre-training: The paper's first contribution is scaling SAC for pre-training. However, it fails to cite work that specifically discusses the *challenges* of scaling SAC in massively parallel simulators and its typical instability compared to PPO like [2]. (b) Model-Based Fine-Tuning Landscape: The paper's core contribution is a model-based fine-tuning method, but it omits citing some other model-based adaptation strategies, such as learning residual "delta action" models (e.g., ASAP [3]) or other concurrent humanoid world-model papers (e.g., Hu et al., 2025 [4]). I believe a whole paragraph of this line of related work is needed.

References:

[1] Hu, Kaizhe, et al. "Robot Trains Robot: Automatic Real-World Policy Adaptation and Learning for Humanoids." Conference on Robot Learning (CoRL). 2025.

[2] Raffin, Antonin. "Getting SAC to Work on a Massive Parallel Simulator..." Blog Post. 2025.

[3] He, Tairan, et al. "ASAP: Aligning Simulation and Real-World Physics for Learning Agile Humanoid Whole-Body Skills." Robotics: Science and Systems (RSS). 2025.

[4] Xinyang Gu, et al. "Learning Humanoid Locomotion with World Model Reconstruction." Robotics: Science and Systems (RSS). 2024.

**Questions:**

The major questions are addressed in the weakness section. Here are some additional questions:

*  What are the primary practical barriers that prevented the authors from attempting the stage (iii) *fine-tuning* loop on the real robot? How would you propose to manage the safety of the initial "deterministic" data collection in the real world?
* In stage (iii), the world model and the policy are fine-tuned concurrently (Sec 4.3). How are these updates interleaved? Is there a risk of the policy overfitting to an inaccurate, partially-fine-tuned world model, and how is this instability managed (e.g., how is the world model rollout horizon scheduled)?
* In Section 4.1, the paper states it relies on SAC's state-dependent variance for exploration. How critical was this choice during pre-training compared to a simpler state-independent variance? Did this choice have any downstream effects on the fine-tuning stage (e.g., better exploration *within* the world model)?

---

> ### Comment · Reviewer_2CYj · 2025-11-23
>
> As the discussion period comes near an end, I hope the authors could address the reviewers' concerns.

---

> > ### Author Response · Authors · 2025-11-23
> > **Thank You for the Reminder**
> >
> > Thank you for the follow-up. We have been running the real-robot fine-tuning experiments during the discussion period. These experiments are now complete, and we will upload an updated version of the paper with the new results as soon as possible.

---

> ### Author Response · Authors · 2025-11-26
> **Official Reply to Reviewer 2CYj (1/3)**
>
> Thank you very much for your thorough review and valuable suggestions. In response to each of your points, we have provided detailed replies and made corresponding adjustments to our paper.
>
> > (Weakness #1) Lack of Real-World Fine-Tuning (Sim-to-Sim): The paper's primary weakness is that its core claim—safe, efficient fine-tuning—is only validated in a "sim-to-sim" setting (MuJoCo to Brax). While the paper shows zero-shot pre-training on a real robot (Appendix A.6), it does not "close the loop" and test the fine-tuning procedure in the real world. The claim of "safety" is significantly weaker without this, as collecting even "deterministic" data in a new real-world environment carries risks (e.g., from initial state distribution mismatch or model error) that the paper does not address.
>
> During the rebuttal phase, we conducted real-world fine-tuning experiments on the Booster T1 humanoid robot. We first pretrained a policy in the MuJoCo-Playground T1LowDimJoystickFlatTerrain task, where we removed most energy-related constraint terms and kept only the action-rate $L_2$ penalty. Although this policy transfers well between simulators (MuJoCo -> Brax), the reduced energy constraints and flat-terrain pretraining lead to a failure in zero-shot sim-to-real transfer.
>
> We then used this policy as the initialization for real-world fine-tuning to evaluate the effectiveness of our method. A video demonstration is available on our [project website](https://lift-humanoid.github.io/#:~:text=Real%2Dworld%20Finetuning%20in%20Booster%20T1). As shown in the video, the policy improves steadily as more real-world data are collected: the reward increases, and the executed velocity becomes more stable and higher. After collecting 80–590 s of data, the robot shows a more upright posture, smoother gait patterns, and more stable forward velocity. These results demonstrate that our fine-tuning framework can successfully adapt a weak sim-to-real policy and make it substantially more robust after collecting only several minutes of data.
>
> We acknowledge that collecting deterministic trajectories in the real world carries inherent risks due to model errors in the actor network. To ensure safety, our real-world experiments include strict termination conditions: an episode is immediately stopped whenever the base height, base linear velocity, base angular velocity, base Euler angles, joint positions, or joint velocities exceed predefined thresholds. Human operators also remotely terminate any episode in which the robot displays unsafe behavior, such as moving toward obstacles.
> Once termination is triggered, we stop the policy’s control and switch the Booster T1 into damping mode, which immediately restricts joint motion to prevent further risk. Afterward, we use the platform’s default walking controller to guide the robot back to the starting area, followed by a default standing controller to reset it to a preparation state. This procedure maintains consistent initial joint positions and velocities across episodes, helping to mitigate the effects of initial-state distribution mismatch during data collection.
>
> We've included all those real-world fine-tuning results and discussions in the revised manuscript Section 5.2 (Finetuning Experiments) and Section 6 (Conclusion and Discussion). Some video demos can be seen in: [https://lift-humanoid.github.io/#:~:text=Real%2Dworld%20Finetuning%20in%20Booster%20T1](https://lift-humanoid.github.io/#:~:text=Real%2Dworld%20Finetuning%20in%20Booster%20T1)
>
>
> > (Weakness #2)  Missing Discussion on Safety for Real-World Adaptation: Following W1, the paper overlooks the practical hardware challenges of data collection for fine-tuning. Even a deterministic policy can fail. Methods like RTR [1] have shown that specially designed safety systems (e.g., a "teacher" robot arm) are crucial for enabling this kind of real-world adaptation on humanoids. A discussion of these practical requirements is missing.
>
> We thank the reviewer for pointing out this related work and have added the corresponding citation and discussion in Section 6 (Conclusion and Discussion). The “teacher” robot arm used in RTR is indeed an effective mechanism to enhance safety and automate resets for smaller platforms. However, to scale this approach to human-sized humanoids, as the robot grows larger, the assisting arm must become substantially more powerful, making the additional hardware difficult to deploy, maintain, and generalize to real-world or in-the-wild environments.

---

> ### Author Response · Authors · 2025-11-26
> **Official Reply to Reviewer 2CYj (2/3)**
>
> > (Weakness #3) WIncomplete Related Work and Missing Citations: The related work section is sparse in several key areas, weakening the paper's positioning and claims of novelty. (a) Context for SAC Pre-training: The paper's first contribution is scaling SAC for pre-training. However, it fails to cite work that specifically discusses the challenges of scaling SAC in massively parallel simulators and its typical instability compared to PPO like [2]. (b) Model-Based Fine-Tuning Landscape: The paper's core contribution is a model-based fine-tuning method, but it omits citing some other model-based adaptation strategies, such as learning residual "delta action" models (e.g., ASAP [3]) or other concurrent humanoid world-model papers (e.g., Hu et al., 2025 [4]). I believe a whole paragraph of this line of related work is needed.
>
> We thank the reviewer for highlighting these missing references. In the revised version, we have substantially expanded the Related Work section to explicitly address this line of work:
>
> (1) Add Raffin's work on scaling SAC to the end of "Sample-Efficient RL";
>
> (2) Retitle "Physics-Informed Neural Networks" to "Model-Based Techniques", and add discussions on prior works including mentioned [3][4].
>
> (3) But for the paper "Hu et al., 2025[4]", we only find the following article with different author names, and we included it as reference in our revised manuscript,
>
> Sun, Wandong, et al. "Learning humanoid locomotion with world model reconstruction." arXiv preprint arXiv:2502.16230 (2025).
>
> > (Question #1)  What are the primary practical barriers that prevented the authors from attempting the stage (iii) fine-tuning loop on the real robot? How would you propose to manage the safety of the initial "deterministic" data collection in the real world?
>
> We thank the reviewer for raising this point. In the revised manuscript we now include hardware fine-tuning experiments on Booster T1. However, several practical constraints still exist:
>
> (1) Our method relies on base-height estimation for both world-model training and reward computation, but Booster T1 does not provide this onboard, so we must use a Vicon motion-capture system. This limits the robot to a small tracking area and requires frequent human supervision to keep it within view;
>
> (2) We estimate base linear velocity by integrating IMU acceleration, which introduces
> drift and may limit policy tracking performance.
>
> (3) Each fine-tuning iteration currently runs sequentially—data collection to 8 seconds at 50 Hz, world-model updates, then synthetic rollouts for policy updates—so even experiments that use only minutes of real data take many hours of wall-clock time. The robot must remain powered on between iterations, leading to frequent battery swaps and multi-hour recharge cycles.
>
> We view these as engineering rather than conceptual constraints, and expect that adopting an asynchronous pipeline similar to SERL[1], together with camera-based height and velocity estimation onboard, would make repeated real-world finetuning substantially more practical.
>
> Regarding safety during the initial deterministic real-world data collection, we propose two **safety protocols** as follows:
>
> **1. Safety checks and human supervision.**
> We monitor the robot state at frequencies above 50~Hz and immediately terminate an episode if the base height, joint positions, joint velocities, or other safety-critical variables exceed predefined limits (see Weakness 1). Human operators also intervene to stop the robot whenever unsafe behavior is observed.
>
> **2. Short episodes during initial fine-tuning reduce risk.**
> Even with a deterministic policy, unsafe actions may occur early in fine-tuning, causing episodes to terminate quickly (typically within 1--2 seconds). We find that even these short trajectories provide meaningful gradients: both the world model and the policy improve significantly after incorporating them. In future work, an asynchronous fine-tuning framework---similar to SERL[1]---could further increase the update-to-data (UTD) ratio, improving safety by collecting short trajectories data for training during data collection.
>
> [1] Jianlan Luo et al., ``SERL: A Software Suite for Sample-Efficient Robotic Reinforcement Learning.''

---

> ### Author Response · Authors · 2025-11-26
> **Official Reply to Reviewer 2CYj (3/3)**
>
> > (Question #2) In stage (iii), the world model and the policy are fine-tuned concurrently (Sec 4.3). How are these updates interleaved? Is there a risk of the policy overfitting to an inaccurate, partially-fine-tuned world model, and how is this instability managed (e.g., how is the world model rollout horizon scheduled)?
>
> We first fine-tune the world model and then freeze both the world model and the actor to generate synthetic rollouts, which are used to fine-tune the actor–critic. We then regenerate data using the updated actor and the (still) frozen world model and fine-tune the actor–critic again. This cycle of world-model-based data generation and actor–critic fine-tuning is repeated $num_{\mathrm{train}} = 1000$ times after the world model fine-tuning. We now include pseudocode in Appendix B to detail the full pipeline.
>
> There is indeed a risk that the policy overfits to an inaccurate or only partially fine-tuned world model. A widely used practical trick to mitigate this is to inject random noise into the states generated by the world model before feeding them to the policy. This technique was already observed in the classical World Models paper (see Table 2 in [2]): as the injected noise increases, the policy’s reward in the world model typically decreases, while its reward in the actual environment improves. Subsequent model-based RL methods such as MBPO [3] and Dreamer [4] also adopt this idea.
>
> In our experiments, we likewise find that sampling next states from the Gaussian predicted by the world model (rather than using the mean deterministically) is important in practice: without this, the policy’s actual reward fails to improve for some seeds. We mention this implementation detail in Section 4.2, but we do not claim it as a contribution of our work. In addition, we linearly increase the world model rollout horizon $H_{wm}$ and the number of actor–critic updates $num_{\mathrm{train}}$ during fine-tuning to further mitigate overfitting, as shown in the pseudocode in Appendix B, but this schedule has only a limited effect on overall performance in practice although we adpoted it.
>
> > (Question # 3) In Section 4.1, the paper states it relies on SAC's state-dependent variance for exploration. How critical was this choice during pre-training compared to a simpler state-independent variance? Did this choice have any downstream effects on the fine-tuning stage (e.g., better exploration within the world model)?
>
> We do not observe a clear advantage of SAC’s state-dependent variance for pretraining exploration in this paper. However, we choose the state-dependent variance because it provides more stable and effective exploration when training inside the world model. The corresponding discussion and experiments are provided in Appendix A.1 (Preliminary Study).
>
> In brief, with a state-independent variance, the initial action standard deviation (std) has a strong impact on the distribution of states explored in the world model. A larger std makes training difficult to converge and tends to drive the policy into regions where the world model is poorly trained, while a smaller std leads to under-exploration and unstable convergence, with some seeds failing to learn at all. In contrast, using a state-dependent variance that outputs both the mean and std of the action distribution substantially improves stability, likely because the variance can adapt to the state and learn how to explore within the world model. This observation is consistent with common practice in model-based RL, where state-dependent exploration is typically used in the world model [3–4]. We therefore adopt state-dependent variance for both pretraining and fine-tuning.
>
>
> References:
>
> [2] Ha D, Schmidhuber J. World models[J]. arXiv preprint arXiv:1803.10122, 2018, 2(3).
>
> [3] Janner M, Fu J, Zhang M, et al. When to trust your model: Model-based policy optimization[J]. Advances in neural information processing systems, 2019, 32.
>
> [4] Hafner D, Pasukonis J, Ba J, et al. Mastering diverse domains through world models[J]. arXiv preprint arXiv:2301.04104, 2023.

---

### Author Response · Authors · 2025-11-26
**General Comments**

We sincerely thank all reviewers and address all reviewer comments in detail below. We have incorporated the corresponding improvements into the revised manuscript. The revised version includes the following major updates (with modified sections clearly marked in red in the manuscript):

1. Real-world fine-tuning experiments on the Booster T1 humanoid have been added to Section 5.2 and Appendix A.3.  (from reviewer 2CYj, JXM1).

2. We revised the Introduction for clarity, improved motivation for physics-informed models, and removed ambiguous or misleading statements highlighted by the reviewers.  (from reviewer 2rhQ)

3. The Related Work section has been expanded to discuss scaling SAC, delta-action model approaches (e.g., ASAP), recent humanoid world-model papers. (from reviewer 2CYj)

4. We moved the training curve of the pretraining stage to Appendix A.2 for saving space and moved the training curve of the ablation studies (physics-informed vs non-physics-informed models, effect of pretraining) into the main paper (Section 5.3). (from reviewer 2rhQ)

5. The SSRL baseline has been integrated into Section 5.2. (from reviewer 2rhQ)

6. We added an extended discussion on non-locomotion tasks (at whole body motion tracking preliminary experiments),  vision-input (DreamerV3), and safety mechanisms such as RTR in Appendix A.7 and Section 6. (from reviewer 2CYj, JXM1, 2rhQ)

7. We included pseudocode detailing the fine-tuning pipeline in Appendix B for transparency and reproducibility. (from reviewer 2CYj)

8. All experimental videos—including **real-robot fine-tuning**, **whole-body tracking** have been uploaded to our project website: [https://lift-humanoid.github.io](https://lift-humanoid.github.io) (We would like to kindly request your patience—the project website may take longer to load at the moment because it hosts a large number of videos.)

---

### Meta-Review · Area_Chair_7bXM · 2025-12-29

**Summary:**

The paper “Towards Bridging the Gap between Large-Scale Pretraining and Efficient Finetuning for Humanoid Control” addresses the challenge of combining scalable humanoid policy pretraining with safe and sample-efficient adaptation to new environments. The authors propose LIFT, a three-stage framework that (1) scales off-policy SAC to large-scale parallel simulation for efficient humanoid pretraining, (2) learns a physics-informed world model from pretraining data, and (3) performs safe fine-tuning by executing deterministic policies in the real environment while confining stochastic exploration to the learned model. Experiments on Unitree G1 and Booster T1 humanoids demonstrate competitive pretraining performance and stable adaptation to out-of-distribution conditions, highlighting a practical integration of large-scale off-policy learning and model-based fine-tuning for humanoid control.

**Reviewer Concerns:**

The paper’s strengths include a well-integrated end-to-end pipeline that combines scalable off-policy pretraining with model-based fine-tuning, a practical use of physics-informed world models to enable safe adaptation, and convincing demonstrations on real humanoid platforms. Common concerns raised by reviewers include the limited evidence for safe fine-tuning in real-world settings beyond simulation, the restricted scope of evaluation primarily focused on locomotion tasks, and the need for clearer justification and positioning of the physics-informed world model relative to existing model-based and pretraining approaches.

**Reviewer Scores:**

Overall, the authors provided thorough and constructive responses to the reviewers’ questions, addressing key concerns through additional experiments, clearer explanations, and revisions to the presentation. In particular, questions regarding real-world fine-tuning safety, the role and necessity of the physics-informed world model, and comparisons to relevant baselines were answered with concrete clarifications and newly added results. Given the discussion, the scores would have remained the same or improved slightly rather than decreased.

---

### Decision · Program_Chairs · 2026-01-26

Accept (Poster)